# Locally-Adaptive Nonparametric Online Learning

**Ilja Kuzborskij** *
DeepMind
iljak@google.com

**Nicolò Cesa-Bianchi**
Dept. of Computer Science & DSRC
University of Milan, Italy
nicolo.cesa-bianchi@unimi.it

## Abstract

One of the main strengths of online algorithms is their ability to adapt to arbitrary data sequences. This is especially important in nonparametric settings, where performance is measured against rich classes of comparator functions that are able to fit complex environments. Although such hard comparators and complex environments may exhibit local regularities, efficient algorithms, which can provably take advantage of these local patterns, are hardly known. We fill this gap by introducing efficient online algorithms (based on a single versatile master algorithm) each adapting to one of the following regularities: *(i)* local Lipschitzness of the competitor function, *(ii)* local metric dimension of the instance sequence, *(iii)* local performance of the predictor across different regions of the instance space. Extending previous approaches, we design algorithms that dynamically grow hierarchical $\varepsilon$-nets on the instance space whose prunings correspond to different "locality profiles" for the problem at hand. Using a technique based on tree experts, we simultaneously and efficiently compete against all such prunings, and prove regret bounds each scaling with a quantity associated with a different type of local regularity. When competing against "simple" locality profiles, our technique delivers regret bounds that are significantly better than those proven using the previous approach. On the other hand, the time dependence of our bounds is not worse than that obtained by ignoring any local regularities.

## 1 Introduction

In online convex optimization [34, 10], a learner interacts with an unknown environment in a sequence of rounds. In the specific setting considered in this paper, at each round $t = 1, 2, \ldots$ the learner observes an instance $\boldsymbol{x}_t \in \mathcal{X} \subset \mathbb{R}^d$ and outputs a prediction $\widehat{y}_t$ for the label $y_t \in \mathcal{Y}$ associated with the instance. After predicting, the learner incurs the loss $\ell_t(\widehat{y}_t)$. We consider two basic learning problems: regression with square loss, where $\mathcal{Y} \equiv [0, 1]$ and $\ell_t(\widehat{y}_t) = \frac{1}{2}(y_t - \widehat{y}_t)^2$, and binary classification with absolute loss, where $\mathcal{Y} \equiv \{0, 1\}$ and $\ell_t(\widehat{y}_t) = |y_t - \widehat{y}_t|$ (or, equivalently, $\ell_t(\widehat{y}_t) = \mathbb{P}(y_t \neq Y_t)$ for randomized predictions $Y_t$ with $\mathbb{P}(Y_t = 1) = \widehat{y}_t$). The performance of a learner is measured through the notion of *regret*, which is defined as the amount by which the cumulative loss of the learner predicting with $\widehat{y}_1, \widehat{y}_2, \ldots$ exceeds the cumulative loss —on the same sequence of instances and labels— of any function $f$ in a given reference class of functions $\mathcal{F}$. Formally,

$$R_T(f) = \sum_{t=1}^{T} \left( \ell_t(\widehat{y}_t) - \ell_t\big(f(\boldsymbol{x}_t)\big) \right) \qquad \forall f \in \mathcal{F}. \tag{1}$$

In order to capture complex environments, we focus on nonparametric classes $\mathcal{F}$ of Lipschitz functions $f : \mathcal{X} \to \mathcal{Y}$. The specific approach adopted in this paper is inspired by the simple and versatile

algorithm from [11], henceforth denoted with `HM`, achieving a regret bound of the form [2]

$$R_T(f) \stackrel{\mathcal{O}}{=} \begin{cases} (\ln T)(L\,T)^{\frac{d}{d+1}} & \text{(square loss)} \\ L^{\frac{d}{d+2}}\,T^{\frac{d+1}{d+2}} & \text{(absolute loss)} \end{cases} \qquad \forall f \in \mathcal{F}_L \qquad (2)$$

for any given $L > 0$. Here $\mathcal{F}_L$ is the class of $L$-Lipschitz functions $f : \mathcal{X} \to \mathcal{Y}$ such that

$$\left| f(\boldsymbol{x}) - f(\boldsymbol{x}') \right| \le L \, \| \boldsymbol{x} - \boldsymbol{x}' \| \qquad (3)$$

for all $\boldsymbol{x}, \boldsymbol{x}' \in \mathcal{X}$, where $\mathcal{X}, \mathcal{Y}$ are compact.[3] Although Lipschitzness is a standard assumption in nonparametric learning, a function in $\mathcal{F}_L$ may alternate regions of low variation with regions of high variation. This implies that, if computed locally (i.e., on pairs $\boldsymbol{x}, \boldsymbol{x}'$ that belong to the same small region), the value of the smallest $L$ satisfying (3) would change significantly across these regions. If we knew in advance the local Lipschitzness profile, we could design algorithms that exploit this information to gain a better control on regret.

Although, for $d \ge 2$, asymptotic rates $T^{(d-1)/d}$ improving on (2) can be obtained using different and more complicated algorithms [4], it is not clear whether these other algorithms can be made locally adaptive in a principled way as we do with `HM`.

**Local Lipschitzness.** Our first contribution is an algorithm for regression with square loss that competes against all functions in $\mathcal{F}_L$. However, unlike the regret bound (2) achieved by `HM`, the regret $R_T(f)$ of our algorithm depends in a detailed way on the local Lipschitzness profile of $f$. Our algorithm operates by sequentially constructing a $D$-level hierarchical $\varepsilon$-net $\mathcal{T}$ of the instance space $\mathcal{X}$ with balls whose radius $\varepsilon$ decreases with each level of the hierarchy. The $D$ levels are associated with local Lipschitz constants $L_1 < L_2 < \cdots < L_D = L$, all provided as an input parameter to the algorithm.

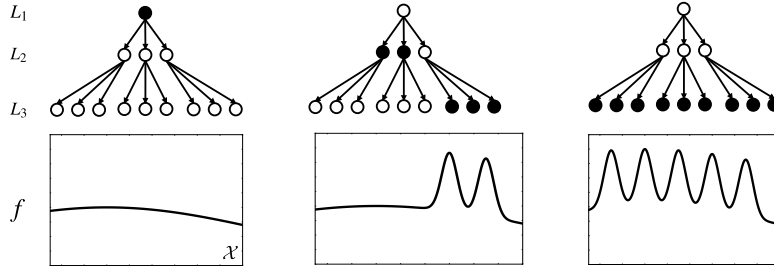

Figure 1: Matching functions to prunings. Profiles of local smoothness correspond to prunings so that smoother functions are matched to smaller prunings.

If we view the hierarchical net as a $D$-level tree whose nodes are the balls in the net at each level, then the local Lipschitzness profile of a function $f$ translates into a pruning of this tree (this is visually explained in Figure 1). By training a local predictor in each ball, we can use the leaves of a pruning $E$ to approximate a function whose local Lipschitz profile "matches" $E$. Namely, a function that satisfies (3) with $L = L_k$ for all observed instances $\boldsymbol{x}, \boldsymbol{x}'$ that belong to some leaf of $E$ at level $k$, for all levels $k$ (since $E$ is a pruning of the hierarchical net $\mathcal{T}$, there is a one-to-one mapping between instances $\boldsymbol{x}_t$ and leaves of $E$). Because our algorithm is simultaneously competitive against *all* prunings, it is also competitive against all functions whose local Lipschitz profile —with respect to the instance sequence— is matched by some pruning. More specifically, we prove that for any $f \in \mathcal{F}_L$ and for any pruning $E$ matching $f$ on the sequence $\boldsymbol{x}_1, \ldots, \boldsymbol{x}_T$ of instances,

$$R_T(f) \stackrel{\widetilde{\mathcal{O}}}{=} \mathbb{E}\left[ L_K^{\frac{d}{d+1}} \right] T^{\frac{d}{d+1}} + \sum_{k=1}^{D} (L_k\,T_{E,k})^{\frac{d}{d+1}} \qquad (4)$$

where, from now on, $T_{E,k}$ always denotes the total number of time steps $t$ in which the current instance $\boldsymbol{x}_t$ belongs to a leaf at level $k$ of the pruning $E$. The expectation is with respect to the random variable $K$ that takes value $k$ with probability equal to the fraction of leaves of $E$ at level $k$. The first term in the right-hand side of (4) bounds the estimation error, and is large when most of the *leaves* of $E$ reside at deep levels (i.e., $f$ has just a few regions of low variation). The second term bounds the approximation error, and is large whenever most of the *instances* $\boldsymbol{x}_t$ belongs to leaves of $E$ at deep levels.

In order to compare this bound to (2), consider $L_k = 2^k$ with $L = L_D = 2^D$. If $f$ is matched by some pruning $E$ such that most instances $\boldsymbol{x}_t$ belong to shallow leaves of $E$, then our bound on $R_T(f)$ becomes of order $T^{d/(d+1)}$, as opposed to the bound of (2) which is of order $(2^D T)^{d/(d+1)}$. On the other hand, for any $f \in \mathcal{F}_L$ we have at least a pruning matching the function: the one whose leaves are all at the deepest level of tree. In this case, our bound on $R_T(f)$ becomes of order $(2^D T)^{d/(d+1)}$, which is asymptotically equivalent to (2). This shows that, up to log factors, our bound is never worse than (2), and can be much better in certain cases. Figure 2 shows this empirically in a toy one-dimensional case.

Our locally adaptive approach can be generalized beyond Lipschitzness. Next, we present two additional contributions where we show that variants of our algorithm can be made adaptive with respect to different local properties of the problem.

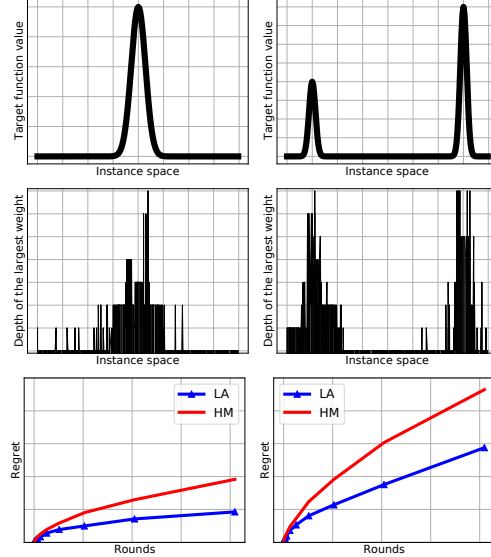

Figure 2: The first row shows two target functions with different Lipschitz profiles. The second row shows the best pruning found by our algorithm, expressed using the depth of the largest weights along each tree-path. The last row show the regret of our algorithm (LA) compared to that of HM, which is given the true Lipschitz constant.

**Local dimension.** It is well known that non-parametric regret bounds inevitably depend exponentially on the metric dimension of the set of data points [11, 27]. Similarly to local Lipschitzness, we want to take advantage of cases in which most of the data points live on manifolds that locally have a low metric dimension. In order to achieve a dependence on the "local dimension profile" in the regret bound, we propose a slight modification of our algorithm, where each level $k$ of the hierarchical $\varepsilon$-net is associated with a local dimension bound $d_k$ such that $d = d_1 > \cdots > d_D$. Note that —unlike local Lipschitzness— the local dimension is *decreasing* as the tree gets deeper. This happens because higher-dimensional balls occupy a larger volume than lower-dimensional ones with the same radius, and so they occur at shallower levels of the tree.

We say that a pruning of the tree associated with the hierarchical $\varepsilon$-net matches a sequence $\boldsymbol{x}_1, \ldots, \boldsymbol{x}_T$ of instances if the number of leaves of the pruning at each level $k$ is $\mathcal{O}\big((L\,T)^{d_k/(1+d_k)}\big)$. For regression with square loss we can prove that, for any $f \in \mathcal{F}_L$ and for any pruning $E$ matching $\boldsymbol{x}_1, \ldots, \boldsymbol{x}_T$, this modified algorithm achieves regret

$$R_T(f) \stackrel{\widetilde{\mathcal{O}}}{=} \mathbb{E}\left[(L\,T)^{\frac{d_K}{1+d_K}}\right] + \sum_{k=1}^{D} (L\,T_{E,k})^{\frac{d_k}{1+d_k}} \tag{5}$$

where, as before, the expectation is with respect to the random variable $K$ that takes value $k$ with probability equal to the fraction of leaves of $E$ at level $k$. If most $\boldsymbol{x}_t$ lie in a low-dimensional manifold of $\mathcal{X}$, so that $\boldsymbol{x}_1, \ldots, \boldsymbol{x}_T$ is matched by some pruning $E$ with deeper leaves, we obtain a regret of order $(L\,T)^{d_D/(1+d_D)}$. This is nearly a parametric rate whenever $d_D \ll d$. In the worst case, when all instances are concentrated at the top level of the tree, we still recover (2).

**Local loss bounds.** Whereas the local Lipschitz profile measures a property of a function with respect to an instance sequence, and the local dimension profile measures a property of the instance

sequence, we now consider the *local loss profile*, which measures a property of a base online learner with respect to a sequence of examples $(\boldsymbol{x}_t, y_t)$. The local loss profile describes how the cumulative losses of the local learners at each node (which are instances of the base online learner) change across different regions of the instance space. To this end, we introduce the functions $\tau_k$, which upper bound the total loss incurred by the local learners at level $k$. We can use the local learners on the leaves of a pruning $E$ to predict a sequence of examples whose local loss profile matches that of $E$. By matching we mean that the local learners run on the subsequence of examples $(\boldsymbol{x}_t, y_t)$ belonging to leaves at level $k$ of $E$ incur a total loss bounded by $\tau_k(T_{E,k})$, for all levels $k$. In order to take advantage of good local loss profiles, we focus on losses —such as the absolute loss— for which we can prove "first-order" regret bounds that scale with the loss of the expert against which the regret is measured. For the absolute loss, the algorithm we consider attains regret

$$R_T(f) \overset{\mathcal{O}}{=} \mathbb{E}\left[(L\,\tau_K(T))^{\frac{d}{d+2}}\right] + \sum_{k=1}^{D}(L\,\tau_k(T_{E,k}))^{\frac{d+1}{d+2}} + \sqrt{\mathbb{E}\left[(L\,\tau_K(T))^{\frac{d}{d+2}}\right]\sum_{k=1}^{D}\tau_k(T_{E,k})} \quad (6)$$

for any $f \in \mathcal{F}_L$, where —as before— the expectation is with respect to the random variable $K$ that takes value $k$ with probability equal to the fraction of leaves of $E$ at level $k$. For concreteness, set $\tau_k(n) = n^{\frac{1}{D-k+1}}$, so that deeper levels $k$ correspond to loss rates that grow faster with time. When $E$ has shallow leaves and $T_{E,k}$ is negligible for $k > 1$, the regret becomes of order $(L\,T^{\frac{1}{D}})^{\frac{d+1}{d+2}}$, which has significantly better dependence on $T$ than $L^{\frac{d}{d+2}}T^{\frac{d+1}{d+2}}$ achieved by HM. Note that we always have a pruning matching all sequences: the one whose leaves are all at the deepest level of the tree. Indeed, $\tau_D(n) = n$ is a trivial upper bound on the absolute loss of any online local learner. In this case, our bound on $R_T(f)$ becomes of order $(L\,T)^{\frac{d+1}{d+2}}$, which is asymptotically equivalent in $T$ compared to (2). Note that our dependence on the Lipschitz constant is slightly worse than (2). This happens because we have to pay an extra constant term for the regret in each ball, which is unavoidable in any first-order regret bound.

**Intuition about the proof.** HM greedily constructs a net on the instance space, where each node hosts a local online learner and the label for a new instance is predicted by the learner in the nearest node. Balls shrinking at polynomial rate are centered on each node, and a new node is created at an instance whenever that instance falls outside the union of all current balls. The algorithms we present here generalize this approach to a hierarchical construction of $\varepsilon$-nets at multiple levels. Each ball at a given level contains a lower-level $\varepsilon$-net using balls of smaller radius, and we view this nested structure of nets as a tree. Radii are now tuned not only with respect to time, but also with respect to the level $k$, where the dependence on $k$ is characterized by the specific locality setting (i.e., local smoothness, local dimension, or local losses). The main novelty of our proof is the fact that we analyze HM in a level-wise manner, while simultaneously competing against the best pruning over the entire hierarchy. Our approach is adaptive because the regret now depends on both the number of leaves of the best pruning and on the number of observations made by the pruning at each level. In other words, if the best pruning has no leaves at a particular level, or is active for just a few time steps at that level, then the algorithm will seldom use the local learners hosted at that level.

Our main algorithmic technology is the sleeping experts framework from [8], where the local learner at each node is viewed as an expert, and active (non-sleeping) experts at a given time step are those along the root-to-leaf path associated with the current instance. For regression with square loss, we use exponential weights (up to re-normalization due to active experts). For classification with absolute loss, we avoid the tuning problem by resorting to a parameter-free algorithm (specifically, we use AdaNormalHedge of [19] although other approaches could work as well). This makes our approach computationally efficient: despite the exponential number of experts in the comparison class, we only pay in the regret a factor corresponding to the depth of the tree.

All omitted proofs can be found in the supplementary material.

## 2 Definitions

Throughout the paper, we assume instances $\boldsymbol{x}_t$ have a bounded arbitrary norm, $\|\boldsymbol{x}_t\| \le 1$, so that $\mathcal{X}$ is the unit ball with center in $\boldsymbol{0}$. We use $\mathcal{B}(\boldsymbol{z}, r)$ to denote the ball of center $\boldsymbol{z} \in \mathbb{R}^d$ and radius $r > 0$, and we write $\mathcal{B}(r)$ instead of $\mathcal{B}(\boldsymbol{0}, r)$.

**Definition 1** (Coverings and packings). *An $\varepsilon$-cover of a set $\mathcal{X}_0 \subseteq \mathcal{X}$ is a subset $\{\boldsymbol{x}_1', \dots, \boldsymbol{x}_n'\} \subset \mathcal{X}_0$ such that for each $\boldsymbol{x} \in \mathcal{X}_0$ there exists $i \in \{1, \dots, n\}$ such that $\|\boldsymbol{x} - \boldsymbol{x}_i'\| \leq \varepsilon$. An $\varepsilon$-packing of a set $\mathcal{X}_0 \subseteq \mathcal{X}$ is a subset $\{\boldsymbol{x}_1', \dots, \boldsymbol{x}_m'\} \subset \mathcal{X}_0$ such that for any distinct $i, j \in \{1, \dots, m\}$, $\|\boldsymbol{x}_i' - \boldsymbol{x}_j'\| > \varepsilon$. An $\varepsilon$-net of a set $\mathcal{X}_0 \subseteq \mathcal{X}$ is any set of points in $\mathcal{X}_0$ which is both an $\varepsilon$-cover and an $\varepsilon$-packing.*

**Definition 2** (Metric dimension). *A set $\mathcal{X}$ has metric dimension $d$ if there exists $C > 0$ such that, for all $\varepsilon > 0$, $\mathcal{X}$ has an $\varepsilon$-cover of size at most $C\,\varepsilon^{-d}$.*

We consider the following online learning protocol with oblivious adversary. Given an unknown sequence $(\boldsymbol{x}_1, y_1), (\boldsymbol{x}_2, y_2), \dots \in \mathcal{X} \times \mathcal{Y}$ of instances and labels, for every round $t = 1, 2, \dots$

1. The environment reveals the instance $\boldsymbol{x}_t \in \mathcal{X}$.
2. The learner selects an action $\widehat{y}_t \in \mathcal{Y}$ and incurs the loss $\ell(\widehat{y}_t, y_t)$.
3. The learner observes $y_t$.

In the rest of the paper, we use $\ell_t(\widehat{y}_t)$ as an abbreviation for $\ell(\widehat{y}_t, y_t)$.

**Hierarchical nets, trees, and prunings.** A *pruning* of a rooted tree is the tree obtained after the application of zero or more replace operations, where each replace operation deletes the subtree rooted at an internal node without deleting the node itself (which becomes a leaf).

Recall that our algorithms work by sequentially building a hierarchical net of the instance sequence. This tree-like structure is defined as follows.

**Definition 3** (Hierarchical net). *A hierarchical net of depth $D$ of an instance sequence $\boldsymbol{\sigma}_T = (\boldsymbol{x}_1, \dots, \boldsymbol{x}_T)$ is a sequence of nonempty subsets[4] $S_1 \subset \cdots \subset S_D \subseteq \{1, \dots, T\}$ and radii $\varepsilon_1 > \cdots > \varepsilon_D > 0$ satisfying the following property: For each level $k = 1, \dots, D$, the set $S_k$ is a $\varepsilon_k$-net of the elements of $\boldsymbol{\sigma}_T$ with balls $\{\mathcal{B}(\boldsymbol{x}_s, \varepsilon_k)\}_{s \in S_k}$.*

Any such hierarchical net can be viewed as a rooted tree $\mathcal{T}$ (conventionally, the root of the tree is the unit ball $\mathcal{X}$, i.e., $S_0 = \{0\}$, $\boldsymbol{x}_0 = \boldsymbol{0}$ and $\varepsilon_0 = 1$) defined by the parent function, where $\boldsymbol{x}_s = \mathrm{PARENT}(\boldsymbol{x}_t)$, if $\boldsymbol{x}_t \in \mathcal{B}(\boldsymbol{x}_s, \varepsilon_k)$ for $s \in S_k$ (if there are more $s$ such that $\boldsymbol{x}_t \in \mathcal{B}(\boldsymbol{x}_s, \varepsilon_k)$, then take the smallest one), while $t \in S_{k+1}$ and $k = 0, 1, \dots, D - 1$. Given an instance sequence $\boldsymbol{\sigma}_T$, let $\boldsymbol{\mathcal{T}}_D(\boldsymbol{\sigma}_T)$ be the family of all trees $\mathcal{T}$ of depth $D$ generated from $\boldsymbol{\sigma}_T$ by choosing the $\varepsilon_k$-nets at each level in all possible ways given a fixed sequence $\{\varepsilon_k\}_{k=1}^D$.

Given $\mathcal{T}$ and a pruning $E$ of $\mathcal{T}$, we use $\mathrm{LEAVES}_k(\mathcal{T}, E)$ to denote the subset of $S_k$ containing the nodes of $\mathcal{T}$ that correspond to leaves of $E$. When $\mathcal{T}$ is clear from the context, we abbreviate $\mathrm{LEAVES}_k(\mathcal{T}, E)$ with $E_k$. For any fixed $\mathcal{T} \in \boldsymbol{\mathcal{T}}_D(\boldsymbol{\sigma}_T)$ let also $|E| = |E_1| + \cdots + |E_D|$ be the number of leaves in $E$.

## 3 Related work

In nonparametric prediction, a classical topic in statistics, one is interested in predicting well compared to the best function in a large class, which typically includes all functions with certain regularities. While standard approaches assume uniform regularity of the optimal function (such as Lipschitzness or Hölder continuity), local minimax rates for adaptive estimation have been studied for nearly thirty years [2, 7, 18] and several works have investigated nonparametric regression under local smoothness assumptions [20, 28].

The nonstochastic setting of nonparametric prediction was investigated by [30, 31, 32], who analyzed the regret of algorithms against Lipschitz function classes with bounded metric entropy. Later, [26] used a non-constructive argument to establish minimax regret rates $T^{(d-1)/d}$ (when $d > 2$) for both square and absolute loss. Inspired by their work, [9] devised the first online algorithms for nonparametric regression enjoying minimax regret. In this work, we employ a nested packing approach, which bears a superficial resemblance to the construction of [9] and to the analysis technique of [26]. However, the crucial difference is that we hierarchically cover the input space, rather than the function class, and use local no-regret learners within each element of the cover. Our algorithm is conceptually similar to the one of [11], however their space packing can be viewed as a "flat" version of the one proposed here, while their analysis only holds for a known time horizon (see also [16] for extensions). Our algorithms adapt to the regularity of the problem in an online

fashion using the tree-expert variant [12] of prediction with expert advice —see also [5]. In this setting, there is a tree-expert for each pruning of a complete tree with a given branching factor. Although the number of such prunings is exponential, predictions and updates can be performed in time linear in the tree depth $D$ using the context tree algorithm of [33]. In this work, we consider a conceptually simpler version, which relies on *sleeping experts* [8]. The goal is to compete against the best pruning in hindsight, which typically requires knowledge of the pruning size for tuning purposes. In case of prediction with absolute loss, we avoid the tuning problem by exploiting a parameter-free algorithm. Local adaptivity to regularities of a competitor, as discussed in the current paper, can be also viewed as automatic parameter tuning through hierarchical expert advice. A similar idea, albeit without the use of a hierarchy, was explored by [29] for automatic step size tuning in online convex optimization —see [25] for a detailed discussion on the topic. Adaptivity of $k$-NN regression and kernel regression to the local effective dimension of the stochastic data-generating process was studied by [14, 15], however they considered a notion of locality different from the one studied here. The idea of adaptivity to the global effective dimension, combined with the net construction of [11] in the online setting, were proposed by [16]. [17] investigated a stronger form of adaptivity to the dimension in nonparametric online learning. Finally, adaptivity to local Lipschitzness was also explored in optimization literature [21, 23].

## 4  Description of the algorithm

Recall that we identify a hierarchical net $S_1, \ldots, S_D$ with a tree $\mathcal{T}$ whose nodes correspond to the elements of the net. Our algorithm predicts using a $\mathcal{T}$ evolving with time, and competes against the best pruning of the tree corresponding to the final hierarchical net. A local online learner is associated with each node of $\mathcal{T}$ except for the root. When a new instance $\boldsymbol{x}_t$ is observed, it is matched with a center $\boldsymbol{x}_s \in S_k$ at each level $k$ (which could be $\boldsymbol{x}_t$ itself, if a new ball is created in the net) until a leaf is reached. The local learners associated with these centers output predictions, which are then aggregated using an algorithm for prediction with expert advice where the local learner at each node is viewed as an expert. Since only a fraction of experts (i.e., those associated with the matched centers, which form a path in a tree) are active at any given round, this can be viewed as an instance of the "sleeping experts" framework of [8]. In the regression case, since the square loss is exp-concave for bounded predictions, we can directly apply the results of [8]. In the classification case, instead, we use a parameter-free approach.

One might wonder whether our dynamically evolving net construction could be replaced by a fixed partition of the instance space chosen at the beginning. As this fixed partition would depend on the time horizon, we would need to use a cumbersome doubling trick to periodically re-start the algorithm from scratch. Moreover, identifying the elements of the partition could be computationally challenging for certain choices of the underlying metric. On the other hand, our algorithm is locally adaptive in any metric space, and does not require the knowledge of the time horizon.

Algorithm 1 contains the pseudocode for the case of exp-concave loss functions. As input, the algorithm requires a radius-tuning function $\rho(k, t)$ which provides the radius of balls at level $k$ and time $t$ given the local regularity parameters (e.g., $L_1 < L_2 < \cdots < L_D$). In the following, we consider specific application-dependent functions $\rho$. The algorithm invokes two subroutines `propagate` and `update`. The former collects the predictions of the local learners along the path of active experts corresponding to an incoming instance, allocating new balls whenever necessary; the latter updates the active experts. We use $\boldsymbol{\pi}_t$ to denote the root-to-leaf path in $\mathcal{T}$ of active experts associated with the current instance $\boldsymbol{x}_t$. The vector $\boldsymbol{\pi}_t$ is built by the subroutine `propagate` along with the vector $\widehat{\boldsymbol{y}}_t$ of their predictions. Both these vectors are then returned to the algorithm (line 4). The sum $W_{t-1}$ of the current weight $w_{\boldsymbol{v}, t-1}$ of each active expert on the path $\boldsymbol{\pi}_t$ is computed in line 5, where $\boldsymbol{v} \sqsubseteq \boldsymbol{\pi}_t$ is used to denote a node in $\mathcal{T}$ whose path is a prefix of $\boldsymbol{\pi}_t$. This sum is used to compute the aggregated prediction on line 6. After observing the true label $y_t$ (line 7), the subroutine `update` updates the active experts in $\boldsymbol{\pi}_t$. Finally, the weights of the active experts are updated (lines 9 and 10).

We now describe a concrete implementation of `propagate` which will be used in Section 5. For simplicity, we assume that all variables of the meta-algorithm which are not explicitly given as input values are visible. The subroutine `propagate` finds in a tree $\mathcal{T}$ the path of active experts associated with an instance $\boldsymbol{x}_t$. When invoked at time $t = 1$, the tree is created as a list of nested balls with common center $\boldsymbol{x}_1$ and radii $\varepsilon_{k,1}$ for $k = 1, \ldots, D$ (lines 4–5). For all $t > 1$, starting from the root

---

**Algorithm 1** Locally Adaptive Online Learning (Hedge style)

---

**Require:** Depth parameter $D$, radius tuning function $\rho : \mathbb{N} \times \mathbb{N} \mapsto \mathbb{R}$

1:   $S_1 \leftarrow \varnothing, \ldots, S_D \leftarrow \varnothing$                                               $\triangleright$ Centers at each level

2:   **for** each round $t = 1, 2, \ldots$ **do**

3:       Receive $\boldsymbol{x}_t$                                           $\triangleright$ **Prediction**

4:       $\left(\boldsymbol{\pi}_t, \widehat{\boldsymbol{y}}_t\right) \leftarrow \texttt{propagate}(\boldsymbol{x}_t, t)$                           $\triangleright$ Subroutine 2

5:       $W_{t-1} \leftarrow \displaystyle\sum_{\boldsymbol{v} \sqsubseteq \boldsymbol{\pi}_t} w_{\boldsymbol{v}, t-1}$

6:       Predict $\widehat{y}_t \leftarrow \dfrac{1}{W_{t-1}} \displaystyle\sum_{\boldsymbol{v} \sqsubseteq \boldsymbol{\pi}_t} w_{\boldsymbol{v}, t-1} \widehat{y}_{\boldsymbol{v}, t}$

7:       Observe $y_t$                                                $\triangleright$ **Update**

8:       $\texttt{update}(\boldsymbol{\pi}_t, \boldsymbol{x}_t, y_t)$

9:       $Z_{t-1} \leftarrow \dfrac{1}{W_{t-1}} \displaystyle\sum_{\boldsymbol{v} \sqsubseteq \boldsymbol{\pi}_t} w_{\boldsymbol{v}, t-1} e^{-\frac{1}{2}\ell_t(\widehat{y}_{\boldsymbol{v}, t})}$

10:      For each $\boldsymbol{v} \sqsubseteq \boldsymbol{\pi}_t$, $w_{\boldsymbol{v}, t} \leftarrow \dfrac{1}{Z_{t-1}} w_{\boldsymbol{v}, t-1} e^{-\frac{1}{2}\ell_t(\widehat{y}_{\boldsymbol{v}, t})}$

11: **end for**

---

node set as parent node (line 1), the procedure finds in each level $k$ the center $\boldsymbol{x}_s$ closest to the current instance $\boldsymbol{x}_t$ among those centers which belong to the parent node (line 8). Note that the parent node is a ball and therefore there is at least one center in $\mathcal{B}_{\text{PARENT}}$. If $\boldsymbol{x}_t$ is in the ball with center $\boldsymbol{x}_s$, then the predictor located at $\boldsymbol{x}_s$ becomes active (line 10). Otherwise, a new ball with center $\boldsymbol{x}_t$ is created in the net at that level, and a new active predictor is associated with that ball (line 13). The indices of

---

**Subroutine 2** $\texttt{propagate}$.

---

**Require:** instance $\boldsymbol{x}_t \in \mathcal{X}$, time step index $t$

1:   $\mathcal{B}_{\text{PARENT}} \leftarrow \mathcal{X}$                                             $\triangleright$ Start from root

2:   **for** depth $k = 1, \ldots, D$ **do**

3:       **if** $S_k \equiv \varnothing$ **then**

4:          $S_k \leftarrow \{t\}$                               $\triangleright$ Create initial ball at depth $k$

5:          Create predictor at $\boldsymbol{x}_t$

6:       **end if**

7:       $\varepsilon \leftarrow \rho(k, t)$                                      $\triangleright$ Get current radius

8:       $s \leftarrow \underset{i \in S_k, \, \boldsymbol{x}_i \in \mathcal{B}_{\text{PARENT}}}{\arg\min} \|\boldsymbol{x}_i - \boldsymbol{x}_t\|$         $\triangleright$ Find closest expert at level $k$

9:       **if** $\|\boldsymbol{x}_t - \boldsymbol{x}_s\| \le \varepsilon$ **then**

10:         $\pi_k \leftarrow s$                 $\triangleright$ Closest expert becomes active and is added to path

11:       **else**

12:         $S_k \leftarrow S_k \cup \{t\}$                         $\triangleright$ Add new center to level $k$

13:         Create predictor at $\boldsymbol{x}_t$

14:         $\pi_k \leftarrow t$                     $\triangleright$ New expert becomes active and is added to path

15:       **end if**

16:       $\widehat{y}_{\pi_k, t} \leftarrow$ prediction of active expert         $\triangleright$ Add prediction to prediction vector

17:       $\mathcal{B}_{\text{PARENT}} \leftarrow \mathcal{B}(\boldsymbol{x}_s, \varepsilon)$       $\triangleright$ Set ball of active expert as current element in the net

18: **end for**

**Ensure:** path $\boldsymbol{\pi}$ of active experts and vector $\widehat{\boldsymbol{y}}$ of active expert predictions

---

active predictors are collected in a vector $\boldsymbol{\pi}$, while their predictions are stored in a vector $\widehat{\boldsymbol{y}}$ and then aggregated using Algorithm 1. We use $T_i$ to denote the subset of time steps on which the expert at node $i$ is active. These are the $t \in \{1, \ldots, T\}$ such that $i$ occurs in $\boldsymbol{\pi}_t$.

## 5   Applications

**Local Lipschitzness.** We first consider the case of local Lipschitz bounds for regression with square loss $\ell_t(\widehat{y}) = \frac{1}{2}(y_t - \widehat{y})^2$, where $y_t \in [0, 1]$ for all $t \ge 1$. Here we use Follow-the-Leader (FTL) as

local online predictor. As explained in the introduction, we need to match prunings to functions with certain local Lipschitz profiles. This is implemented by the following definition.

**Definition 4** (Functions admissible with respect to a pruning). *Given $0 < L_1 < \cdots < L_D$, a hierarchical net $\mathcal{T} \in \mathcal{T}_D(\boldsymbol{\sigma}_T)$ of an instance sequence $\boldsymbol{\sigma}_T$, and a time-dependent radius tuning function $\rho$, we define the set of admissible functions with respect to a pruning $E$ of $\mathcal{T}$ by*

$$
\mathcal{F}(E, \mathcal{T}) \equiv \Big\{ f : \mathcal{X} \to [0, 1] \,\Big|\, \forall \boldsymbol{x} \in \mathcal{B}\big(\boldsymbol{x}_i, \rho(k, t)\big), \forall i \in \text{LEAVES}_k(\mathcal{T}, E)
$$
$$
\big|f(\boldsymbol{x}_i) - f(\boldsymbol{x})\big| \le L_k\, \rho(k, t), \quad k = 1, \ldots, D,\ t = 1, \ldots, T \Big\}.
$$

Now we establish a regret bound with respect to admissible functions. Recall that $T_{E,k}$ is the total number of time steps $t$ in which the current instance $\boldsymbol{x}_t$ belongs to a leaf at level $k$ of the pruning $E$.

**Theorem 1.** *Given $0 < L_1 < \cdots < L_D$, suppose that Algorithm 1 using Subroutine 2 is run for $T$ rounds with radius tuning function $\rho(k, t) = (L_k t)^{-\frac{1}{d+1}}$, and let $\mathcal{T}$ be the resulting hierarchical net. Then, for all prunings $E$ of $\mathcal{T}$ the regret $R_T(f)$ satisfies*

$$
R_T(f) \stackrel{\widetilde{\mathcal{O}}}{=} \mathbb{E}\left[L_K^{\frac{d}{d+1}}\right] T^{\frac{d}{d+1}} + \sum_{k=1}^{D} (L_k T_{E,k})^{\frac{d}{d+1}} \qquad \forall f \in \mathcal{F}(E, \mathcal{T}) \,.
$$

The expectation is understood with respect to the random variable $K$ that takes value $k$ with probability equal to the fraction of leaves of $E$ at level $k$.

The prunings $E$ and the admissible functions $\mathcal{F}(E, \mathcal{T})$ depend on the structure of $\mathcal{T}$. In turn, this structure depends on the instance sequence $\boldsymbol{\sigma}_T$ only (except for the analysis of local losses, where it also depends on the local learners). Importantly, *the structure of $\mathcal{T}$*, and therefore the comparator class used in our analyses, *is not determined by the predictions of the algorithm*, a fact that would compromise the definition of regret.

**Local dimension.** We now look at a different notion of adaptivity, and demonstrate that Algorithm 1 is also capable of adapting to the *local* dimension of the data sequence. We consider a decreasing sequence $d = d_1 > \cdots > d_D$ of local dimension bounds, where $d_k$ is assigned to the level $k$ of the hierarchical net maintained by Algorithm 1. We also make a small modification to Subroutine 2. Namely, we add a new center at level $k$ only if the designated size of the net (which depends on the local dimension bound) has not been exceeded. The modified subroutine is `propagateDim`.

---

**Subroutine 3** `propagateDim`.

**Require:** instance $\boldsymbol{x}_t \in \mathcal{X}$, time step index $t$, $C$ (see Def. 2)
1: $\mathcal{B}_{\text{PARENT}} \leftarrow \mathcal{X}$ &emsp;&emsp;&emsp;&emsp;&emsp;&emsp;&emsp;&emsp;&emsp;&emsp;&emsp;&emsp;&emsp; ▷ Start from root
2: **for** depth $k = 1, \ldots, D$ **do**
3: &emsp;&emsp; **if** $S_k \equiv \varnothing$ **then**
4: &emsp;&emsp;&emsp;&emsp; $S_k \leftarrow \{t\}$ &emsp;&emsp;&emsp;&emsp;&emsp;&emsp;&emsp;&emsp;&emsp;&emsp; ▷ Create initial ball at depth $k$
5: &emsp;&emsp;&emsp;&emsp; Create predictor at $\boldsymbol{x}_t$
6: &emsp;&emsp; **end if**
7: &emsp;&emsp; $\varepsilon \leftarrow \rho(k, t)$ &emsp;&emsp;&emsp;&emsp;&emsp;&emsp;&emsp;&emsp;&emsp;&emsp;&emsp; ▷ Get current radius
8: &emsp;&emsp; $s \leftarrow \underset{\substack{i \in S_k \\ \boldsymbol{x}_i \in \mathcal{B}_{\text{PARENT}}}}{\arg\min} \|\boldsymbol{x}_i - \boldsymbol{x}_t\|$ &emsp;&emsp;&emsp; ▷ Find closest expert at level $k$
9: &emsp;&emsp; **if** $|S_k| > C\varepsilon^{d_k}$ **or** $\|\boldsymbol{x}_t - \boldsymbol{x}_s\| \le \varepsilon$ **then**
10: &emsp;&emsp;&emsp;&emsp; $\pi_k \leftarrow s$ &emsp;&emsp;&emsp;&emsp;&emsp;&emsp; ▷ Closest expert becomes active and is added to path
11: &emsp;&emsp; **else if** $|S_k| \le C\varepsilon^{d_k}$ **then**
12: &emsp;&emsp;&emsp;&emsp; $S_k \leftarrow S_k \cup \{t\}$ &emsp;&emsp;&emsp;&emsp;&emsp;&emsp; ▷ Add new center to level $k$
13: &emsp;&emsp;&emsp;&emsp; Create predictor at $\boldsymbol{x}_t$
14: &emsp;&emsp;&emsp;&emsp; $\pi_k \leftarrow t$ &emsp;&emsp;&emsp;&emsp;&emsp;&emsp; ▷ New expert becomes active and is added to path
15: &emsp;&emsp; **end if**
16: &emsp;&emsp; $\widehat{y}_{\pi_k, t} \leftarrow$ prediction of active expert &emsp;&emsp; ▷ Add prediction to prediction vector
17: &emsp;&emsp; $\mathcal{B}_{\text{PARENT}} \leftarrow \mathcal{B}(\boldsymbol{x}_s, \varepsilon)$ &emsp;&emsp; ▷ Set ball of active expert as current element in the net
18: **end for**
**Ensure:** path $\boldsymbol{\pi}$ of active experts and vector $\widehat{\boldsymbol{y}}$ of active expert predictions

---

Since the local dimension assumption is made on the instance sequence rather than on the function class, in this scenario we may afford to compete against the class $\mathcal{F}_L$ of all $L$-Lipschitz functions, while we restrict the prunings to those that are compatible with the local dimension bounds w.r.t. the hierarchical net built by the algorithm.

**Definition 5** (Prunings admissible w.r.t. local dimension bounds)**.** *Given* $d = d_1 > \cdots > d_D$ *and a hierarchical net* $\mathcal{T} \in \boldsymbol{\mathcal{T}}_D(\boldsymbol{\sigma}_T)$ *of an instance sequence* $\boldsymbol{\sigma}_T$, *define the set of admissible prunings by*

$$\mathcal{E}_{\dim}(\mathcal{T}) \equiv \left\{ E \in \mathcal{T} \,:\, \left| \text{LEAVES}_k(\mathcal{T}, E) \right| \leq C \, (L\,T)^{\frac{d_k}{1+d_k}}, \; k = 1, \dots, D \right\}.$$

**Theorem 2.** *Given* $d = d_1 > \cdots > d_D$, *suppose that Algorithm 1 using Subroutine 2 is run for* $T$ *rounds with radius tuning function* $\rho(k, t) = (Lt)^{-\frac{1}{1+d_k}}$, *and let* $\mathcal{T}$ *the resulting hierarchical net. Then, for all prunings* $E \in \mathcal{E}_{\dim}(\mathcal{T})$ *the regret satisfies*

$$R_T(f) \stackrel{\widetilde{\mathcal{O}}}{=} \mathbb{E}\left[ (L\,T)^{\frac{d_K}{1+d_K}} \right] + \sum_{k=1}^{D} (L\,T_{E,k})^{\frac{d_k}{1+d_k}} \qquad \forall f \in \mathcal{F}_L.$$

**Local loss bounds.** The third notion of adaptivity we study is with respect to the loss of the local learners in each node of a hierarchical net. The local loss profile is parameterized with respect to a sequence $\tau_1, \dots, \tau_D$ of nonnegative and nondecreasing $\tau_k : \{1, \dots, T\} \to \mathbb{R}$ such that each $\tau_k$ bounds the total loss of all local learners at level $k$ of the hierarchical net. In order to achieve better regrets when the data sequence can be predicted well by local learners in a shallow pruning we assume $\tau_1(n) < \cdots < \tau_D(n) = n$ for all $n = 1, \dots, T$, where the choice of $\tau_D(n) = n$ allows us to fall back to the standard regret bounds if the data sequence is hard to predict.

Unlike our previous applications, focused on regression with the square loss, we now consider binary classification with absolute loss $\ell_t(\widehat{y}_t) = |\widehat{y}_t - y_t|$, which —unlike the square loss— is not exp-concave. As we explained in Section 1, using losses that are not exp-concave is motivated by the presence of first-order regret bounds, which allow us to take advantage of good local loss profiles. While the exp-concavity of the square loss dispensed us from the need of tuning Algorithm 1 using properties of the pruning, here we circumvent the tuning issue by replacing Algorithm 1 with the parameter-free Algorithm 4 (stated in Appendix A.1), which is based on the AdaNormalHedge algorithm of [19]. As online local learners we use self-confident Weighted Majority [5, Exercise 2.10] with two constant experts predicting 0 and 1. In the following, we denote by $\Lambda_{i,T}$ the cumulative loss of a local learner at node $i$ over the time steps $T_i$ when the expert is active. Similarly to the previous section, we compete against the class $\mathcal{F}_L$ of all Lipschitz functions, and introduce the following:

$$\mathcal{E}_{\text{loss}}(\mathcal{T}) \equiv \left\{ E \in \mathcal{T} \,:\, \sum\nolimits_{i \in \text{LEAVES}_k(\mathcal{T}, E)} \Lambda_{i,T} \leq \tau_k(T_{E,k}), \; k = 1, \dots, D \right\}.$$

If $E \in \mathcal{E}_{\text{loss}}(\mathcal{T})$, the total loss of all the leaves at a particular level behaves in accordance with $(\tau)_{i=1}^{D}$.

**Theorem 3.** *Suppose that the Algorithm 4 runs self-confident weighted majority at each node with radius tuning function* $\rho(k, t) = (L\tau_k(t))^{-\frac{1}{2+d}}$ *and let* $\mathcal{T}$ *the resulting hierarchical net. Then for all pruning* $E \in \mathcal{E}_{\text{loss}}(\mathcal{T})$ *and for all* $f \in \mathcal{F}_L$ *the regret satisfies*

$$R_T(f) \stackrel{\widetilde{\mathcal{O}}}{=} \mathbb{E}\left[ (L\,\tau_K(T))^{\frac{d}{2+d}} \right] + \sum_{k=1}^{D} (L\,\tau_k(T_{E,k}))^{\frac{1+d}{2+d}} + \sqrt{ \mathbb{E}\left[ (L\,\tau_K(T))^{\frac{d}{2+d}} \right] \sum_{k=1}^{D} \tau_k(T_{E,k}) }.$$

## 6 Future work

Our algorithm, based on prediction with tree experts, is computationally efficient: the running time at each step is only logarithmic in the size of the tree. On the other hand, because the algorithm constructs the tree dynamically, adding a new path of size $\mathcal{O}(D)$ in each round, space grows linearly in time (note that the algorithm never allocates the entire tree, but only the paths corresponding to active experts). An interesting avenue for future research is to investigate extensions of our algorithm to *bounded* space prediction models, similarly to other online nonparametric predictors, e.g., budgeted kernelized Perceptron [3, 6]. Beside regression, we also presented a locally-adaptive version of our algorithm for randomized binary classification through absolute loss. Our proofs can be easily extended to any exp-concave loss functions, as these do not require any tuning of learning rates in local predictors (tuning local learning rates would complicate our analysis). We also believe that it is possible to extend our approach to any convex loss through a parameter-free local learner, such as those proposed in [13, 24].

# 7 Broader impact

We believe that presented research should be categorized as basic research and we are not targeting any specific application area. Theorems may inspire new algorithms and theoretical investigation. The algorithms presented here can be used for many different applications and a particular use may have both positive or negative impacts. We are not aware of any immediate short term negative implications of this research and we believe that a broader impact statement is not required for this paper.

## Acknowledgments and Disclosure of Funding

We are grateful to András György, Sébastien Gerchinovitz, and Pierre Gaillard for many insightful comments. Nicolò Cesa-Bianchi is partially supported by the MIUR PRIN grant Algorithms, Games, and Digital Markets (ALGADIMAR) and by the EU Horizon 2020 ICT-48 research and innovation action under grant agreement 951847, project ELISE (European Learning and Intelligent Systems Excellence).

## Footnotes

*Work partly done while at the University of Milan, Italy.

[2]We use $f \stackrel{\mathcal{O}}{=} g$ to denote $f = \mathcal{O}(g)$ and $f \stackrel{\widetilde{\mathcal{O}}}{=} g$ to denote $f = \widetilde{\mathcal{O}}(g)$.

[3]The bound for the square loss, which is not contained in [11], can be proven with a straightforward extension of the analysis in that paper.

[4] Here the net $S_k$ is defined using the indices $s$ of the points $\boldsymbol{x}_s$.

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
