[Supplementary Material]

# Locally-Adaptive Nonparametric Online Learning: Supplementary Material

**Ilja Kuzborskij**
DeepMind
iljak@google.com

**Nicolò Cesa-Bianchi**
Dept. of Computer Science & DSRC
University of Milan, Italy
nicolo.cesa-bianchi@unimi.it

## A  Omitted algorithms

### A.1  Algorithm for nonparametric classification with local losses

Instead of the standard exponential weights on which the updates of Algorithm 1 are based, AdaNormalHedge performs update using the function

$$\psi(r,c) = \frac{1}{2}\left(\exp\left(\frac{[r+1]_+^2}{3(c+1)}\right) - \exp\left(\frac{[r-1]_+^2}{3(c+1)}\right)\right) \ .$$

---

**Algorithm 4** Locally Adaptive Online Learning (AdaNormalHedge style)

---

**Require:** Depth parameter $D$, radius tuning function $\rho : \mathbb{N} \times \mathbb{N} \mapsto \mathbb{R}$
1:  $S_1 \leftarrow \varnothing, \dots, S_D \leftarrow \varnothing$                              ▷ Centers at each level
2:  **for** each round $t = 1, 2, \dots$ **do**
3:      Receive $\boldsymbol{x}_t$                                                                ▷ **Prediction**
4:      $(\boldsymbol{\pi}_t, \widehat{\boldsymbol{y}}_t) \leftarrow \texttt{propagate}(\boldsymbol{x}_t, t)$    ▷ Algorithm 2
5:      **for** each $\boldsymbol{v} \sqsubseteq \boldsymbol{\pi}_t$ **do**
6:          **if** $t = 1$ **then**
7:              $w_{\boldsymbol{v},t} \leftarrow \psi(0,0)$
8:          **else**
9:              $w_{\boldsymbol{v},t} \leftarrow \psi(\bar{r}_{\boldsymbol{v},t-1}, C_{\boldsymbol{v},t-1})$
10:         **end if**
11:     **end for**
12:     Predict   $\widehat{y}_t \leftarrow \dfrac{1}{Z_t} \sum_{\boldsymbol{v} \sqsubseteq \boldsymbol{\pi}_t} w_{\boldsymbol{v},t}\, \widehat{y}_{\boldsymbol{v},t}$   where   $Z_t = \sum_{\boldsymbol{v} \sqsubseteq \boldsymbol{\pi}_t} w_{\boldsymbol{v},t}$
13:     Observe $y_t$                                                                            ▷ **Update**
14:     $\texttt{update}(\boldsymbol{\pi}_t, \boldsymbol{x}_t, y_t)$
15:     $\bar{\ell}_t \leftarrow \sum_{\boldsymbol{v} \sqsubseteq \boldsymbol{\pi}_t} w_{\boldsymbol{v},t}\ell_t(\widehat{y}_{\boldsymbol{v},t})$
16:     **for** each $\boldsymbol{v} \sqsubseteq \boldsymbol{\pi}_t$ **do**
17:         $r_{\boldsymbol{v},t} \leftarrow \bar{\ell}_t - \ell_t(\widehat{y}_{\boldsymbol{v},t}), \quad \bar{r}_{\boldsymbol{v},t} \leftarrow \bar{r}_{\boldsymbol{v},t-1} + r_{\boldsymbol{v},t}, \quad C_{\boldsymbol{v},t} \leftarrow C_{\boldsymbol{v},t-1} + |r_{\boldsymbol{v},t}|$
18:     **end for**
19: **end for**

---

## B  Learning with expert advice over trees

In order to prove the regret bounds in our locally-adaptive learning setting, we start by deriving bounds for prediction with expert advice when the competitor class is all the prunings of a tree whose each node hosts an expert, a framework initially investigated by [12]. Our analysis uses the sleeping

experts setting of [8], in which only a subset $\mathcal{E}_t$ of the node experts are active at each time step $t$. In our locally-adaptive setting, the set of active experts at time $t$ corresponds to the active root-to-leaf path $\boldsymbol{\pi}_t$ selected by the current instance $\boldsymbol{x}_t$ —see Section 4. The inactive experts at time $t$ neither output predictions nor get updated. The prediction of a pruning $E$ at time $t$, denoted with $f_{E,t}$ is the prediction $\widehat{y}_{i,t}$ of the node expert corresponding to the unique leaf $i$ of $E$ on $\boldsymbol{\pi}_t$.

---

**Algorithm 5** Learning over trees through sleeping experts

---
**Require:** Tree $\mathcal{T}$ and initial weights for each node of the tree
1: **for** each round $t = 1, 2, \ldots$ **do**
2:     Observe predictions of active experts $\mathcal{E}_t$ (corresponding to a root-to-leaf path in the tree)
3:     Predict $\widehat{y}_t$ and observe $y_t$
4:     Update the weight of each active expert
5: **end for**

---

Next, we consider two algorithms for the problem of prediction with expert advice over trees. In order to be simultaneously competitive with all prunings, we need algorithms that do not require tuning of their parameters depending on the specific pruning against which the regret is measured. In case of exp-concave losses (like the square loss) tuning is not required and Hedge-style algorithms work well. In case of generic convex losses, we use the more complex parameterless algorithm AdaNormalHedge.

We start by recalling the algorithm for learning with sleeping experts and the basic regret bound of [8]. The sleeping experts setting assumes a set of $M$ experts without any special structure. At every time step $t$ only an adversarially chosen subset $\mathcal{E}_t$ of the experts provides predictions and gets updated —see Algorithm 6. The regret bound is parameterized in terms of the relative entropy $\mathrm{KL}(\boldsymbol{u} \,||\, \boldsymbol{w}_1)$

---

**Algorithm 6** Exponential weights with sleeping experts for $\eta$-exp-concave losses

---
**Require:** Initial nonnegative weights $\{w_{i,1}\}_{i=1,\ldots,M}$
1: **for** each round $t = 1, 2, \ldots$ **do**
2:     Receive predictions $\widehat{y}_{i,t}$ of active experts $i \in \mathcal{E}_t$
3: $\qquad \widehat{y}_t = \dfrac{\sum_{i \in \mathcal{E}_t} w_{i,t}\, \widehat{y}_{i,t}}{\sum_{i \in \mathcal{E}_t} w_{i,t}}$ $\qquad\qquad\qquad\qquad\qquad\qquad\qquad\qquad$ ▷ Prediction
4:     Observe $y_t$
5: $\quad$ For $i \in \mathcal{E}_t \quad w_{i,t+1} = \dfrac{w_{i,t}\, e^{-\eta \ell_t(\widehat{y}_{i,t})}}{\sum_{j \in \mathcal{E}_t} w_{j,t}\, e^{-\eta \ell_t(\widehat{y}_{j,t})}} \sum_{j \in \mathcal{E}_t} w_{j,t}$ $\qquad\qquad$ ▷ Update
6: **end for**

---

between the initial of distribution over experts $\boldsymbol{w}_1$ and any target distribution $\boldsymbol{u}$. The following theorem states a slightly more general bound that holds for any $\eta$-exp-concave loss function (for completeness, the proof is given in Appendix D).

**Theorem 4** ([8]). *If Algorithm 6 is run on any sequence $\ell_1, \ldots, \ell_T$ of $\eta$-exp-concave loss functions, then for any sequence $\mathcal{E}_1, \ldots, \mathcal{E}_T \subseteq \{1, \ldots, M\}$ of awake experts and for any distribution $\boldsymbol{u}$ over $\{1, \ldots, M\}$, the following holds*

$$\sum_{t=1}^{T} U_t\, \ell_t(\widehat{y}_t) - \sum_{t=1}^{T} \sum_{i \in \mathcal{E}_t} u_i\, \ell_t(\widehat{y}_{i,t}) \le \frac{1}{\eta}\, \mathrm{KL}\left( \boldsymbol{u} \,\middle\|\, \frac{\boldsymbol{w}_1}{\|\boldsymbol{w}\|_1} \right) \tag{7}$$

*where* $U_t = \sum_{i \in \mathcal{E}_t} u_i$.

By taking $\boldsymbol{w}_1$ to be uniform over the experts, the above theorem implies a bound with a $\ln M$ factor. However, since we predict and perform updates only with respect to *awake* experts, this can be improved to $\ln M_T$, where $M_T$ is the number of distinct experts ever awake throughout the $T$ time steps. The following lemma (whose proof is deferred to Appendix D) formally states this fact.

Fix a sequence $\mathcal{E}_1, \ldots, \mathcal{E}_T \subseteq \{1, \ldots, M\}$ of awake experts such that $\left| \mathcal{E}_1 \cup \cdots \cup \mathcal{E}_T \right| = M_T$. Let the uniform distribution supported over the awake experts, denoted with $\boldsymbol{w}_1^{\mathcal{E}}$, be defined by $w_{i,1}^{\mathcal{E}} = 1/M_T$ if $i \in \mathcal{E}_1 \cup \cdots \cup \mathcal{E}_T$ and 0 otherwise.

**Lemma 1.** *Suppose Algorithm 6 is run with initial weights $w_{i,1} = 1$ for $i = 1, \ldots, M$ and with a sequence $\mathcal{E}_1, \ldots, \mathcal{E}_T \subseteq \{1, \ldots, M\}$ of awake experts. Then the regret of the algorithm initialized with $\boldsymbol{w}_1$ matches the regret of the algorithm initialized with $\boldsymbol{w}_1^{\mathcal{E}}$.*

We use Theorem 4 and Lemma 1 to derive a regret bound for Algorithm 5 when predictions and updates are provided by Algorithm 6. The same regret bound can be achieved through the analysis of [22, Theorem 3], albeit their proof follows a different argument.

**Theorem 5.** *Suppose that Algorithm 5 is run using predictions and updates provided by Algorithm 6. Then, for any sequence $\ell_1, \ldots, \ell_T$ of $\eta$-exp-concave losses and for any pruning $E$ of the input tree $\mathcal{T}$,*

$$\sum_{t=1}^{T} \left( \ell_t(\widehat{y}_t) - \ell_t(f_{E,t}) \right) \leq \frac{|E|}{\eta} \ln \frac{M_T}{|E|} \ .$$

*Proof.* Let $\boldsymbol{u}$ be the uniform distribution over the $|E|$ terminal nodes of $E$. At each round, exactly one terminal node of $E$ is in the active path of $\mathcal{T}$. Therefore $\ell_t(f_{E,t}) = \sum_{i \in \mathcal{E}_t} u_i \ell_t(\widehat{y}_{i,t})$, and also $U_t = \frac{1}{|E|}$ for all $t$ because only one expert in $\mathcal{E}_t$ is awake in the support of $\boldsymbol{u}$. Now note that although the algorithm is actually initialized with $w_{1,i} = 1$, Lemma 1 shows that the regret remains the same if we assume the algorithm is initialized with $\boldsymbol{w}_1^{\mathcal{E}}$. The choice of the competitor $\boldsymbol{u}$ gives us $\mathrm{KL}(\boldsymbol{u} \,\|\, \boldsymbol{w}_1^{\mathcal{E}}) = \ln \left( M_T/|E| \right)$. By applying Theorem 4 we finally get

$$\sum_{t=1}^{T} U_t \ell_t(\widehat{y}_t) - \sum_{t=1}^{T} \sum_{i \in E_t} u_i \ell_t(\widehat{y}_{i,t})$$

$$= \frac{1}{|E|} \sum_{t=1}^{T} \left( \ell_t(\widehat{y}_t) - \ell_t(f_{E,t}) \right) \qquad \text{(only one expert awake in the active path)}$$

$$\leq \frac{1}{\eta} \ln \frac{M_T}{|E|}$$

concluding the proof. $\qquad\qquad\square$

In case of general convex losses, we simply apply the following theorem where $\Lambda_E = \ell_1(f_{E,1}) + \cdots + \ell_T(f_{E,T})$ is the cumulative loss of pruning $E$.

**Theorem 6** (Section 6 in [19]). *Suppose that Algorithm 5 is run using predictions and updates provided by AdaNormalHedge. Then, for any sequence $\ell_1, \ldots, \ell_T$ of convex losses and for any pruning $E$ of the input tree $\mathcal{T}$,*

$$\sum_{t=1}^{T} \left( \ell_t(\widehat{y}_t) - \ell_t(f_{E,t}) \right) \overset{\widetilde{\mathcal{O}}}{=} \sqrt{|E| \Lambda_E \ln \frac{M_T}{|E|}} \ .$$

## C  Proofs for nonparametric prediction

We start by proving a master regret bound that can be specialized to various settings of interest. Recall that the prediction of a pruning $E$ at time $t$ is $f_{E,t} = \widehat{y}_{i,t}$, where $\widehat{y}_{i,t}$ is the prediction of the node expert sitting at the unique leaf $i$ of the pruning $E$ on the active path $\boldsymbol{\pi}_t$. Recall also that $\boldsymbol{x}_i$ is the center of the ball in the hierarchical net corresponding to node $i$ in the tree. As in our locally-adaptive setting node experts are local learners, $\widehat{y}_{i,t}$ should be viewed as the prediction of the local online learning algorithm sitting at node $i$ of the tree. Let $T_i$ be the subset of time steps when $i$ is on the active path $\boldsymbol{\pi}_t$. We now introduce the definitions of regret for the tree expert

$$R_T^{\mathrm{tree}}(E) = \sum_{t=1}^{T} \left( \ell_t(\widehat{y}_t) - \ell_t(f_{E,t}) \right)$$

and for node expert $i$

$$R_{i,T}^{\mathrm{loc}} = \sum_{t \in T_i} \left( \ell_t(\widehat{y}_{i,t}) - \ell_t(y_i^\star) \right)$$

where $\mathcal{H}$ is either $[0,1]$ (regression with square loss) or $\{0,1\}$ (classification with absolute loss), and

$$y_i^\star = \arg\min_{y \in \mathcal{H}} \sum_{t \in T_i} \ell_t(y) \ .$$

Note that, for all $f : \mathcal{X} \to [0,1]$ and for $y_i^\star$ defined as above,

$$\sum_{t \in T_i} \left( \ell_t(y_i^\star) - \ell_t\big(f(\boldsymbol{x}_i)\big) \right) \le 0 \ . \tag{8}$$

**Lemma 2.** *Suppose that Algorithm 1 (or, equivalently, Algorithm 4) is run on a sequence $\ell_1,\ldots,\ell_T$ of convex and $L'$-Lipschitz losses and let $\mathcal{T}$ be the resulting hierarchical net. Then for any pruning $E$ of $\mathcal{T}$ and for any $f : \mathcal{X} \to \mathcal{Y}$,*

$$R_T(f) \le R_T^{\mathrm{tree}}(E) + \sum_{k=1}^{D} \sum_{i \in \mathrm{LEAVES}_k(E)} R_{T_i}^{\mathrm{loc}} + L' \sum_{k=1}^{D} \sum_{i \in \mathrm{LEAVES}_k(E)} \sum_{t \in T_i} \big| f(\boldsymbol{x}_i) - f(\boldsymbol{x}_t) \big| \ .$$

*Proof.* We decompose regret into two terms: one capturing the regret of the algorithm with respect to a pruning $E$, and one capturing the regret of $E$ against the competitor $f$,

$$R_T(f) = \sum_{t=1}^{T} \left( \ell_t(\widehat{y}_t) - \ell_t\big(f(\boldsymbol{x}_t)\big) \right) = R_T^{\mathrm{tree}}(E) + \sum_{t=1}^{T} \left( \ell_t(f_{E,t}) - \ell_t\big(f(\boldsymbol{x}_t)\big) \right) \ .$$

We now split the second term into estimation and approximation error. Define the prediction of a local learner at node $i$ and time step $t$ as $\widehat{y}_{i,t}$,

$$\sum_{t=1}^{T} \left( \ell_t(f_{E,t}) - \ell_t\big(f(\boldsymbol{x}_t)\big) \right) = \sum_{k=1}^{D} \sum_{i \in \mathrm{LEAVES}_k(E)} \sum_{t \in T_i} \left( \ell_t(\widehat{y}_{i,t}) - \ell_t\big(f(\boldsymbol{x}_t)\big) \right)$$

$$= \sum_{k=1}^{D} \sum_{i \in \mathrm{LEAVES}_k(E)} \sum_{t \in T_i} \left( \ell_t(\widehat{y}_{i,t}) - \ell_t(y_i^\star) \right)$$

$$+ \sum_{k=1}^{D} \sum_{i \in \mathrm{LEAVES}_k(E)} \sum_{t \in T_i} \left( \ell_t(y_i^\star) - \ell_t\big(f(\boldsymbol{x}_t)\big) \right)$$

$$\le \sum_{k=1}^{D} \sum_{i \in \mathrm{LEAVES}_k(E)} R_{i,T}^{\mathrm{loc}} \qquad \text{(regret of local predictors)}$$

$$+ \sum_{k=1}^{D} \sum_{i \in \mathrm{LEAVES}_k(E)} \sum_{t \in T_i} \left( \ell_t\big(f(\boldsymbol{x}_i)\big) - \ell_t\big(f(\boldsymbol{x}_t)\big) \right)$$

$$\le L' \sum_{k=1}^{D} \sum_{i \in \mathrm{LEAVES}_k(E)} \sum_{t \in T_i} \big| f(\boldsymbol{x}_i) - f(\boldsymbol{x}_t) \big|$$

using (8) and the fact that $\ell_t$ is $L'$-Lipschitz. Combining terms completes the proof. $\qquad\square$

The next key lemma bounds the number of leaves in a pruning $E$ for different settings of the ball radius function.

**Lemma 3.** *For any instance sequence $\boldsymbol{\sigma}_T$, for any $\mathcal{T} \in \mathcal{T}_D(\boldsymbol{\sigma}_T)$, and for any pruning $E$ of $\mathcal{T}$, let the random variable $K$ be such that $\mathbb{P}(K = k) = \frac{|E_k|}{|E|}$ for $k = 1,\ldots,D$. Then the following statements hold for each $k$,*

$$|E| \le \mathbb{E}\left[ L_K^{\frac{d}{1+d}} \right] T^{\frac{d}{1+d}} \qquad \text{for} \quad \varepsilon_{k,t} = (L_k t)^{-\frac{1}{1+d}} \qquad \text{(Local Lipschitzness)}$$

$$|E| \le \mathbb{E}\left[ (L T)^{\frac{d_K}{1+d_K}} \right] \qquad \text{for} \quad \varepsilon_{k,t} = (Lt)^{-\frac{1}{1+d_k}} \qquad \text{(Local dimension)}$$

$$|E| \le \mathbb{E}\left[ (L \tau_K(T))^{\frac{d}{2+d}} \right] \qquad \text{for} \quad \varepsilon_{k,t} = (L\tau_k(t))^{-\frac{1}{1+d}} \qquad \text{(Local losses)}$$

*Proof.* We first recall that leaves of a pruning $E$ correspond to balls in a $\varepsilon_{k,T}$-packing. Thus, to give a bound on the number of leaves at level $k$, that is $|E_k|$, we estimate the size of the packing formed at level $k$. However, instead of directly bounding size of the packing, we use a more careful volumetric argument. In particular, at level $k$ w only pack the volume that is not occupied yet by previous levels —this helps to avoid gross overestimates, since we take into account the fact that we can only pack a limited volume. Denote volume of a set in an Euclidean space by $\mathrm{vol}(\cdot)$, and let $\mathrm{pack}_k$ stand for the collection of balls at level $k$ of the packing.

**Local Lipschitzness.** Pick any $k = 1, \ldots, D$. Recalling that $\mathcal{X}$ is the unit ball,

$$|E_k| \leq \frac{\mathrm{vol}(\mathcal{X}) - \mathrm{vol}\left(\bigcup_{s=1}^{k-1} \mathrm{pack}_s\right)}{\mathrm{vol}(\mathcal{B}(\varepsilon_{k,T}))} = \frac{1 - \sum_{s=1}^{k-1} |E_s| \varepsilon_{s,T}^d}{\varepsilon_{k,T}^d}$$

$$= (L_k T)^{\frac{d}{1+d}} - \sum_{s=1}^{k-1} |E_s| \left(\frac{L_k}{L_s}\right)^{\frac{d}{1+d}} \qquad \text{(using the definition of } \varepsilon_{k,t}.\text{)}$$

Dividing both sides by $L_k^{\frac{d}{1+d}}$ we get

$$\sum_{s=1}^{k} \frac{|E_s|}{L_s^{\frac{d}{1+d}}} \leq T^{\frac{d}{1+d}}$$

Since $k$ is chosen arbitrarily, we can set $k = D$ and write

$$\sum_{s=1}^{D} \frac{|E_s|}{L_s^{\frac{d}{1+d}}} \leq T^{\frac{d}{1+d}}$$

or, equivalently,

$$1 \leq \left(\sum_{s=1}^{D} \frac{|E_s|}{L_s^{\frac{d}{1+d}}}\right)^{-1} T^{\frac{d}{1+d}} .$$

Multiplying both sides by $|E|$ gives

$$|E| \leq \left(\sum_{s=1}^{D} \frac{|E_s|/|E|}{L_s^{\frac{d}{1+d}}}\right)^{-1} T^{\frac{d}{1+d}} .$$

Now observe that the factor in the right-hand side is a weighted harmonic mean with weights $\frac{|E_1|}{|E|}, \ldots, \frac{|E_D|}{|E|}$. Therefore the HM-GM-AM inequality (between Harmonic, Geometric, and Arithmetic Mean) implies that

$$|E| \leq \mathbb{E}\left[L_K^{\frac{d}{1+d}}\right] T^{\frac{d}{1+d}}$$

where the expectation is with respect to $\mathbb{P}(K = k) = \frac{|E_k|}{|E|}$ . This proves the first statement.

**Local dimension.** Using again the volumetric argument and the appropriate definition of $\varepsilon_{k,t}$

$$|E_k| \leq \frac{1 - \sum_{s=1}^{k-1} |E_s| \varepsilon_{s,T}^{d_s}}{\varepsilon_{k,T}^{d_k}} = (L T)^{\frac{d_k}{1+d_k}} - \sum_{s=1}^{k-1} |E_s| (L T)^{\frac{d_k}{1+d_k} - \frac{d_s}{1+d_s}} .$$

Dividing both sides by $(L T)^{\frac{d_k}{1+d_k}}$ and rearranging gives

$$|E| \leq \left(\sum_{s=1}^{D} \frac{|E_s|/|E|}{(L T)^{\frac{d_s}{1+d_s}}}\right)^{-1} .$$

Once again, observing that the factor in the right-hand side is a weighted harmonic mean with weights $\frac{|E_1|}{|E|}, \ldots, \frac{|E_D|}{|E|}$, by the HM-GM-AM inequality we get

$$|E| \leq \mathbb{E}\left[(L T)^{\frac{d_K}{1+d_K}}\right]$$

where the expectation is with respect to $\mathbb{P}(K = k) = \frac{|E_k|}{|E|}$.

**Local losses.** Using once more the volumetric argument and the appropriate definition of $\varepsilon_{k,t}$,

$$|E_k| \leq \frac{1 - \sum_{s=1}^{k-1}|E_s|\varepsilon_{s,T}^d}{\varepsilon_{k,T}^d} = (L\,\tau_k(T))^{\frac{d}{2+d}} - \sum_{s=1}^{k-1}|E_s|\left(\frac{\tau_k(T)}{\tau_s(T)}\right)^{\frac{d}{2+d}} .$$

Dividing both sides by $(L\,\tau_k(T))^{\frac{d}{2+d}}$ and multiplying by $|E|$ we get

$$|E| \leq \left(\sum_{s=1}^{D}\frac{|E_s|/|E|}{(L\,\tau_s(T))^{\frac{d}{2+d}}}\right)^{-1} \leq \mathbb{E}\left[(L\,\tau_K(T))^{\frac{d}{2+d}}\right]$$

where —as before— the expectation is with respect to $\mathbb{P}(K=k) = \frac{|E_k|}{|E|}$. The proof is concluded. $\square$

## C.1 Proof of Theorem 1

We start from Lemma 2 with the square loss $\ell_t(y) = \frac{1}{2}(y - y_t)^2$ and $\mathcal{Y} \equiv \mathcal{H} \equiv [0,1]$. As $\ell_t$ is $\eta$-exp-concave for $\eta \leq \frac{1}{2}$ and 1-Lipschitz in $[0,1]$, we can apply Theorem 5 with $L' = 1$. This gives us

$$R_T(f) \leq R_T^{\text{tree}}(E) + \sum_{k=1}^{D}\sum_{i \in \text{LEAVES}_k(E)} R_{i,T}^{\text{loc}} + \sum_{k=1}^{D}\sum_{i \in \text{LEAVES}_k(E)}\sum_{t \in T_i}|f(\boldsymbol{x}_i) - f(\boldsymbol{x}_t)| .$$

Using Theorem 5 combined with $M_T \leq DT$, and then using the first statement of Lemma 3, we get that

$$R_T^{\text{tree}}(E) \overset{\widetilde{\mathcal{O}}}{=} |E| \overset{\widetilde{\mathcal{O}}}{=} \mathbb{E}\left[L_K^{\frac{d}{1+d}}\right]T^{\frac{d}{1+d}} .$$

**Bounding the estimation error.** Using the regret bound of Follow the Leader (FTL) with respect to the square loss [5, p. 43], we get

$$\sum_{k=1}^{D}\sum_{i \in \text{LEAVES}_k(E)} R_{i,T}^{\text{loc}} \leq 8\ln(eT)|E| \leq 8\ln(eT)\,\mathbb{E}\left[L_K^{\frac{d}{1+d}}\right]T^{\frac{d}{1+d}}$$

where we used Lemma 3 to obtain the second inequality.

**Bounding the approximation error.** By hypothesis, $f \in \mathcal{F}(E,\mathcal{T})$. Using Definition 4 and the fact that at time $t$ ball radii at depth $k$ are $\varepsilon_{k,t}$,

$$\sum_{k=1}^{D}\sum_{i \in \text{LEAVES}_k(E)}\sum_{t \in T_i}\big|f(\boldsymbol{x}_i) - f(\boldsymbol{x}_t)\big| \leq \sum_{k=1}^{D}L_k\sum_{i \in \text{LEAVES}_k(E)}\sum_{t \in T_i}\varepsilon_{k,t}$$

$$\leq \sum_{k=1}^{D}L_k\sum_{i \in \text{LEAVES}_k(E)}\sum_{t=1}^{|T_i|}\varepsilon_{k,t}$$

$$= \sum_{k=1}^{D}L_k^{\frac{d}{1+d}}\sum_{i \in \text{LEAVES}_k(E)}\sum_{t=1}^{|T_i|}t^{-\frac{1}{1+d}}$$

$$\leq \sum_{k=1}^{D}L_k^{\frac{d}{1+d}}\int_0^{T_{E,k}}\tau^{-\frac{1}{1+d}}\,\mathrm{d}\tau$$

$$\leq 2\sum_{k=1}^{D}(L_k T_{E,k})^{\frac{d}{1+d}} .$$

Combining the bound on $R_T^{\text{tree}}(E)$ with the bounds on the estimation and approximation errors, we get that

$$R_T(f) \overset{\widetilde{\mathcal{O}}}{=} \mathbb{E}\left[L_K^{\frac{d}{1+d}}\right]T^{\frac{d}{1+d}} + \sum_{k=1}^{D}(L_k T_{E,k})^{\frac{d}{1+d}} \qquad \forall f \in \mathcal{F}(E,\mathcal{T}) \qquad (9)$$

which completes the proof.

## C.2 Proof of Theorem 2

Similarly to the proof of Theorem 1, we use the properties of the square loss and Lemma 2. This gives us

$$R_T(f) \le R_T^{\text{tree}}(E) + \sum_{k=1}^{D} \sum_{i \in \text{LEAVES}_k(E)} R_{i,T}^{\text{loc}} + \sum_{k=1}^{D} \sum_{i \in \text{LEAVES}_k(E)} \sum_{t \in T_i} \left| f(\boldsymbol{x}_i) - f(\boldsymbol{x}_t) \right| .$$

Using Theorem 5 combined with $M_T \le DT$ (the largest number of traversed distinct paths), and then using Lemma 3 (second statement), we get that

$$R_T^{\text{tree}}(E) \overset{\widetilde{\mathcal{O}}}{=} |E| \overset{\widetilde{\mathcal{O}}}{=} \mathbb{E}\left[ (LT)^{\frac{d_K}{1+d_K}} \right] .$$

**Bounding the estimation error.** Using —as before— the regret bound of FTL with respect to the square loss we immediately get

$$\sum_{k=1}^{D} \sum_{i \in \text{LEAVES}_k(E)} R_{T_i}^{\text{loc}} \le 8\ln(eT)|E| \le 8\ln(eT)\,\mathbb{E}\left[ (LT)^{\frac{d_K}{1+d_K}} \right]$$

where the last inequality uses Lemma 3.

**Bounding the approximation error.** For all $f \in \mathcal{F}_L$ and for all $E \in \mathcal{E}_{\dim}(\mathcal{T})$, since at time $t$ the ball radii at depth $k$ are $\varepsilon_{k,t}$,

$$\sum_{k=1}^{D} \sum_{i \in \text{LEAVES}_k(E)} \sum_{t \in T_i} \left| f(\boldsymbol{x}_i) - f(\boldsymbol{x}_t) \right| \le L \sum_{k=1}^{D} \sum_{i \in \text{LEAVES}_k(E)} \sum_{t \in T_i} \varepsilon_{k,t} \tag{10}$$

$$\le L \sum_{k=1}^{D} \sum_{i \in \text{LEAVES}_k(E)} \sum_{t=1}^{|T_i|} \varepsilon_{k,t} \tag{11}$$

$$\le \sum_{k=1}^{D} L^{1-\frac{1}{1+d_k}} \int_{0}^{T_{E,k}} \tau^{-\frac{1}{1+d_k}} \, \mathrm{d}\tau \tag{12}$$

$$\le 2 \sum_{k=1}^{D} (LT_{E,k})^{\frac{d_k}{1+d_k}} . \tag{13}$$

Combining the bound on $R_T^{\text{tree}}(E)$ with the bounds on the estimation and approximation errors, we get that

$$R_T(f) \overset{\widetilde{\mathcal{O}}}{=} \mathbb{E}\left[ (LT)^{\frac{d_K}{1+d_K}} \right] + \sum_{k=1}^{D} (LT_{E,k})^{\frac{d_k}{1+d_k}} \qquad \forall f \in \mathcal{F}_L . \tag{14}$$

The proof is complete.

## C.3 Proof of Theorem 3

Here we use the 1-Lipschitz absolute loss function $\ell_t(y) = |y - y_t|$ and run self-confident Exponentially Weighted Average (EWA) [1] at every node of the tree with $\mathcal{H} \equiv \{0,1\}$. Lemma 2 gives us the decomposition

$$R_T(f) \le R_T^{\text{tree}}(E) + \sum_{k=1}^{D} \sum_{i \in \text{LEAVES}_k(E)} R_{i,T}^{\text{loc}} + \sum_{k=1}^{D} \sum_{i \in \text{LEAVES}_k(E)} \sum_{t \in T_i} \left| f(\boldsymbol{x}_i) - f(\boldsymbol{x}_t) \right| .$$

Theorem 6 gives us

$$R_T^{\text{tree}}(f_E) \overset{\widetilde{\mathcal{O}}}{=} \sqrt{ |E|\Lambda_E \ln\left( \frac{M_T}{|E|} \right) } .$$

Using once more $M_T \leq DT$, the fact that any pruning $E$ has at least one leaf, and Lemma 3 (third statement), we get

$$1 \leq |E| \leq \mathbb{E}\left[\left(L\,\tau_K(T)\right)^{\frac{d}{1+d}}\right] .$$

Recall that $\widehat{y}_{i,t}$ is the output at time $t$ of the local predictor at node $i$. By definition of $\tau_k$,

$$\Lambda_E = \sum_{k=1}^{D} \sum_{i \in \text{LEAVES}_k(E)} \sum_{t \in T_i} \ell_t(\widehat{y}_{i,t}) \leq \sum_{k=1}^{D} \tau_k(T_{E,k}) .$$

This gives us

$$R_T^{\text{tree}}(E) \stackrel{\widetilde{\mathcal{O}}}{=} \sqrt{\left(\sum_{k=1}^{D} \tau_k(T_{E,k})\right) \mathbb{E}\left[\left(L\,\tau_K(T)\right)^{\frac{d}{2+d}}\right]} .$$

**Bounding the estimation error.**  Let the cumulative loss of the best expert for and node $i$ be defined by

$$\Lambda_{i,T}^{\star} = \sum_{t \in T_i} \ell_t(y_i^{\star}) \qquad \text{where} \qquad y_i^{\star} = \arg\min_{y \in \{0,1\}} \sum_{t \in T_i} \ell_t(y)$$

Then, [5, Exercise 2.11] implies that for a positive constant $c$ (independent of the number of experts and $\Lambda_{i,T}^{\star}$), $R_{i,T}^{\text{loc}} \leq 2\sqrt{2\ln(2)\Lambda_{i,T}^{\star}} + c\ln(2)$. We can thus write

$$\sum_{k=1}^{D} \sum_{i \in \text{LEAVES}_k(E)} R_{i,T}^{\text{loc}} \leq \sum_{k=1}^{D} \sum_{i \in \text{LEAVES}_k(E)} \left(2\sqrt{2\ln(2)\Lambda_{i,T}^{\star}} + c\ln(2)\right)$$

$$\leq 2\sqrt{2\ln(2)} \sum_{k=1}^{D} \sqrt{|E_k| \sum_{i \in \text{LEAVES}_k(E)} \Lambda_{i,T}^{\star}} + c\ln(2)|E|$$

$$\leq 2\sqrt{2\ln(2)} \sum_{k=1}^{D} \sqrt{|E_k|\tau_k(T_{E,k})} + c\ln(2)|E|$$

since, according to the definition of $\tau_\kappa$,

$$\sum_{i \in \text{LEAVES}_k(E)} \Lambda_{i,T}^{\star} \leq \tau_k(T_{E,k}) .$$

Next, using the Cauchy-Schwartz inequality,

$$\sum_{k=1}^{D} \sqrt{|E_k|\tau_k(T_{E,k})} \leq \sqrt{\sum_{k=1}^{D} |E_k|} \sqrt{\sum_{k=1}^{D} \tau_k(T_{E,k})} \leq \sqrt{\left(\sum_{k=1}^{D} \tau_k(T_{E,k})\right) \mathbb{E}\left[\left(L\,\tau_K(T)\right)^{\frac{d}{2+d}}\right]}$$

where the last inequality is a consequence of Lemma 3 (third statement). This gives us the following bound on the estimation error

$$\sum_{k=1}^{D} \sum_{i \in \text{LEAVES}_k(E)} R_{i,T}^{\text{loc}} \stackrel{\mathcal{O}}{=} \sqrt{\left(\sum_{k=1}^{D} \tau_k(T_{E,k})\right) \mathbb{E}\left[\left(L\,\tau_K(T)\right)^{\frac{d}{2+d}}\right]} + \mathbb{E}\left[\left(L\,\tau_K(T)\right)^{\frac{d}{2+d}}\right] .$$

**Bounding the approximation error.** Since we are competing against the class of $L$-Lipschitz functions,

$$\sum_{k=1}^{D} \sum_{i \in \text{LEAVES}_k(E)} \sum_{t \in T_i} |f(\boldsymbol{x}_i) - f(\boldsymbol{x}_t)| \leq L \sum_{k=1}^{D} \sum_{i \in \text{LEAVES}_k(E)} \sum_{t \in T_i} \varepsilon_{k,t}$$

$$\leq L \sum_{k=1}^{D} \sum_{i \in \text{LEAVES}_k(E)} \sum_{t=1}^{|T_i|} \varepsilon_{k,t}$$

$$= L^{1-\frac{1}{2+d}} \sum_{k=1}^{D} \sum_{i \in \text{LEAVES}_k(E)} \sum_{t=1}^{|T_i|} \tau_k(t)^{-\frac{1}{2+d}}$$

$$\leq L^{\frac{1+d}{2+d}} \sum_{k=1}^{D} \int_0^{\tau_k(T_{E,k})} \theta^{-\frac{1}{1+d}} \, \mathrm{d}\theta$$

(since $\tau_k$ is non-decreasing)

$$\leq \frac{3}{2} L^{\frac{1+d}{2+d}} \sum_{k=1}^{D} \tau_k(T_{E,k})^{\frac{1+d}{2+d}} \, .$$

Combining all terms together, the final regret bound is

$$R_T(f) \overset{\widetilde{\mathcal{O}}}{=} \sqrt{\left( \sum_{k=1}^{D} \tau_k(T_{E,k}) \right) \mathbb{E}\left[ (L\,\tau_K(T))^{\frac{d}{2+d}} \right] + \mathbb{E}\left[ (L\,\tau_K(T))^{\frac{d}{2+d}} \right] + \sum_{k=1}^{D} (L\,\tau_k(T_{E,k}))^{\frac{1+d}{2+d}}} \, .$$

# D  Additional Proofs

**Proof of Theorem 4.** Recall that by definition of $\eta$-exp-concavity of $\ell_t$, $e^{-\eta \ell_t(x)}$ is concave for all $x$. Observe that the relative entropy satisfies

$$\text{KL}(\boldsymbol{u} \,||\, \boldsymbol{w}_t) - \text{KL}(\boldsymbol{u} \,||\, \boldsymbol{w}_{t+1})$$

$$= \sum_{i=1}^{M} u_i \ln \frac{w_{i,t+1}}{w_{i,t}}$$

$$= \sum_{i \in \mathcal{E}_t} u_i \ln \frac{w_{i,t+1}}{w_{i,t}}$$

$$= -\eta \sum_{i \in \mathcal{E}_t} u_i \, \ell_t(\mu_{i,t}) - U_t \ln \frac{\sum_{j \in \mathcal{E}_t} w_{j,t} \, e^{-\eta \ell_t(\widehat{y}_{j,t})}}{\sum_{j \in \mathcal{E}_t} w_{j,t}}$$  (update step in Alg. 6)

$$\geq -\eta \sum_{i \in \mathcal{E}_t} u_i \, \ell_t(\widehat{y}_{i,t}) + \eta \, U_t \, \ell_t \left( \frac{\sum_{j \in \mathcal{E}_t} w_{j,t} \, \widehat{y}_{j,t}}{\sum_{j \in \mathcal{E}_t} w_{j,t}} \right)$$  (exp-concavity and Jensen's)

$$= -\eta \sum_{i \in \mathcal{E}_t} u_i \, \ell_t(\widehat{y}_{i,t}) + \eta \, U_t \, \ell_t(\widehat{y}_t)$$

Summing both sides over $t = 1, \dots, T$ we get

$$\text{KL}(\boldsymbol{u} \,||\, \boldsymbol{w}_1) \geq \text{KL}(\boldsymbol{u} \,||\, \boldsymbol{w}_1) - \text{KL}(\boldsymbol{u} \,||\, \boldsymbol{w}_T) = -\eta \sum_{t=1}^{T} \sum_{i \in \mathcal{E}_t} u_i \, \ell_t(\widehat{y}_{i,t}) + \eta \sum_{t=1}^{T} U_t \, \ell_t(\widehat{y}_t) \, .$$

The proof is now complete. ∎

**Proof of Lemma 1.** The proof exploits the fact that whenever the weights are initialized uniformly over a subset of the experts, the sequence of predictions remains the same as if the weights were initialized uniformly over all experts. In particular, we show that the predictions obtained assuming

weights are initialized with $w_{i,1} = 1/M_T$ for $i \in \mathcal{E}_1 \cup \cdots \cup \mathcal{E}_T$ with $\left| \mathcal{E}_1 \cup \cdots \cup \mathcal{E}_T \right| = M_T$ are the same as the predictions obtained with $w_{i,1} = 1$ for all $i$. We use an inductive argument to prove that the factor $1/M_T$ introduced by the initialization $w_{i,1} = 1/M_T$ is preserved after each update. Fix a round $t > 1$ and assume that all $w_{i,t-1}$ contain the initialization factor $1/M_T$. Split the set of awake experts into observed ones $\mathcal{E}_t^{\mathrm{o}} \subseteq \mathcal{E}_1 \cup \cdots \cup \mathcal{E}_{t-1}$ (that is experts which were awake at least once before), and unobserved ones $\mathcal{E}_t^{\mathrm{u}} \equiv \mathcal{E}_t \setminus \mathcal{E}_t^{\mathrm{o}}$. Clearly $w_{i,t} = 1/M_T$ for every $i \in \mathcal{E}_t^{\mathrm{u}}$, as they were never updated. For $i \in \mathcal{E}_t^{\mathrm{o}}$, the update rule

$$w_{i,t} = \frac{w_{i,t-1} e^{-\eta \ell_{i,t-1}}}{\sum_{j \in \mathcal{E}_{t-1}} w_{j,t-1} e^{-\eta \ell_{j,t-1}}} \sum_{j \in \mathcal{E}_{t-1}} w_{j,t-1}$$

shows that the initialization factors that occur in the terms $w_{j,t-1}$ contained in the two sums cancel out, whereas the one contained in $w_{i,t-1}$ remains unchanged.

We can now write the prediction at round $t$ as

$$\widehat{y}_t = \frac{\sum_{i \in \mathcal{E}_t} w_{i,t} \, \widehat{y}_{i,t}}{\sum_{i \in \mathcal{E}_t} w_{i,t}} = \frac{\sum_{i \in \mathcal{E}_t^{\mathrm{o}}} w_{i,t} \, \widehat{y}_{i,t} + \sum_{i \in \mathcal{E}_t^{\mathrm{u}}} w_{i,1} \, \widehat{y}_{i,t}}{\sum_{i \in \mathcal{E}_t^{\mathrm{o}}} w_{i,t} + \sum_{i \in \mathcal{E}_t^{\mathrm{u}}} w_{i,1}} = \frac{M_T \sum_{i \in \mathcal{E}_t^{\mathrm{o}}} w_{i,t} \, \widehat{y}_{i,t} + \sum_{i \in \mathcal{E}_t^{\mathrm{u}}} \widehat{y}_{i,t}}{M_T \sum_{i \in \mathcal{E}_t^{\mathrm{o}}} w_{i,t} + |\mathcal{E}_t^{\mathrm{u}}|}$$

$$= \frac{\sum_{i \in \mathcal{E}_t} w'_{i,t} \, \widehat{y}_{i,t}}{\sum_{i \in \mathcal{E}_t} w'_{i,t}}$$

where in the last step we canceled the initialization factor $1/M_T$ from $w_{i,t}$ and introduced $w'_{i,t}$ which differs from $w_{i,t}$ only due to the initialization $w'_{i,1} = 1$. This completes the proof. ∎