[Reviews · NeurIPS 2020]

Review 1

Summary and Contributions: Paper 5218 provides a novel online contextual learning algorithm, that exploits local properties of non-parametric predictors in class F. The document include favorable bounds on the regret, typically depending on the Lipschitz constant and metric dimensionality of F, by exploiting those properties locally. is a general framework, and is applied on the Lipschitz constant, metric dimensionality, and the local loss function to produce three different concrete algorithms with specific bounds. It extends `ALG2` (denoted with HM) in [9], which considers a \epsilon-net over the context space, to include hierarchical levels of decreasing \epsilon (given e.g. local Lipschitz constant). The nodes in this tree host (locally learned) experts. Provided with a new context, the algorithm propagates through the tree, finding the closest previous instance according to epsilon of each level. Their associated experts then become 'active', and their advice is aggregated into a single output and, upon receiving the loss function, are updated (through multiplicative weighting of the active experts, a concept borrowed from "sleeping experts" framework [6]). The authors provide a regret bound on expectation of the *local* property (e.g. Lipschitz constant), which in worst-case within log-factors of previous work (and otherwise favorable). * [9]: Hazan, Elad, and Nimrod Megiddo. “Online Learning with Prior Knowledge.” In International Conference on Computational Learning Theory, 499–513. Springer, 2007. * [6]: Freund, Yoav, Robert E. Schapire, Yoram Singer, and Manfred K. Warmuth. “Using and Combining Predictors That Specialize.” In Proceedings of the Twenty-Ninth Annual ACM Symposium on Theory of Computing, 334–343, 1997.

Strengths: Going beyond algorithms and bounds based on global properties, this work addresses the important concern of exploiting the local landscape. The authors provide a general higher level approach and show how this can be specialized in three concrete methods, accompanied with non-trivial proofs on their respective bounds. Additionally, the method is relatively general and considers both the square and absolute loss, and otherwise supports any arbitrary class of non-parametric prediction functions. The proposed method is implemented and shown, in practice, empirically that it outperforms HM with supporting visualization of the resulting weights of the experts.

Weaknesses: The relevance of this work may be limited to a smaller community as the proposed algorithm has not been tested in practice outside of a 1-dimensional toy problem, and to my best understanding does not have major consequences for researchers not involved with the theory of (local adaptivity in) online contextual learning. The main weakness of the algorithm seems to be the need to specify the hierarchical levels (e.g Lipschitz constants, metric dimensionality) which sound non-trivial. Lastly, the bounds provided by the authors contain expectations which, assuming worst case, (asymptotically) converge to that of previous work. It is not entirely clear whether the additional (algorithm) complexity and input parameters justify the gains, and if so, on what problems.

Correctness: I do not have the qualified understanding or background to verify correctness. The authors have rigorously declared the notation, assumptions and are precise in their statements. In the parts that I was able to understand and follow, I found no faults.

Clarity: The paper is well-written and clear. The proofs seem properly formalized and are accompanied by English descriptions where appropriate. For personal preference, I would recommend to reduce the detail on the method in the description to the "bare" minimal to highlight the contribution. This would leave room to add those details to the algorithm description and applications, and hopefully improve the connection between the concept behind the method, the different local metrics, and the pseudocode.

Relation to Prior Work: This work properly lists the related work and provides a high level comparison with respect to the difference in bounds.

Reproducibility: Yes

Additional Feedback: Comments on readability * In Definition 1: "such that for each x \in X_o there exists i \in...", it might be that you mean "x \in X" * The first sentence of the "Intuition about the proof" is grammatically a little off * It was not entirely clear to me what "estimation error" and "approxmation error" referred to at the end of page 2 ---- Post response ---- Thank you for the response. I am positively surprised to see the only marginal increase in run-time compared to MC. After reading other reviews and your response to them I still believe this is a good submission. ---- Post response ----- Thank you for the response, in particular on clarifying on the role of the hyper-parameters. I claimed the hyper-parameters were a main weakness in my initial review, so my evaluation should have increased as it turns out I was off. However, other reviewers have raised the issue of linear (in time) space requirements. As a result, I will not update my score in general.


Review 2

Summary and Contributions: This paper proposes tree experts based algorithms that can adapt to one of three local patterns/regularities including local Lipschitzness of the competitor function, local metric dimension of the instance space, as well as local performance of the predictor. It extends the previously known tree expert and sleeping expert algorithms to sequentially construct a D-level hierarchical $\epsilon$-net. Each node of the tree is a local online learner represented by a ball with center from sequence and radius as a function of time step t and level k. By using different radius tuning functions, the paper proves that it can adapt to the previously mentioned local patterns, although not simultaneously.

Strengths: Like the authors pointed out in the paper, there are some local pattern adaptive online algorithms. But to the best of my knowledge, the non-parametric setting considered in this paper is new and different from the previous papers' settings. This paper extends the tree experts algorithm and sleeping expert update idea to achieve local pattern adaptivity in non-parametric setting, but its soundness is debatable.

Weaknesses: One of the main concerns I have is how practical the algorithm is. The update time complexity is acceptable, but what about the space/memory complexity? It seems that the algorithm needs to have a pretty large tree with too many nodes to be practical in real applications. In Appendix Lemma 3, it proves the upper bound for the total number of nodes for any pruning E, which is sub-linear in T. But what about the total number of nodes within the tree? Will it increase in exponential order of T or other order? Another limitation is that, based on my understanding, the algorithm can only adapt to one of the mentioned local regularities, and it really just depends on which radius tuning function is used. The statement in the abstract kind of makes me feel it can deal with all of the three cases simultaneously. What's more, the claims are only true with oblivious adversary. This makes the claimed improvement in the local dimension case (which I personally think could be the largest improvement in the upper bound) be doubtful, since the local dimension case only depends on the property of the instance sequence. The oblivious adversary can certainly generate the instance sequence that does not have lower local dimension. Also, the regret bound improvement for local Lipschitzness is not that significant considering how large/complicated the covering net is. The paper assumes that the instance space is the unit ball. What about the case when the instance space is not necessarily a unit ball? Can the algorithm still work in this case? What changes should we make? ----------------after rebuttal------------------- The authors answered some of my concerns, especially the space complexity. Although the linear(T) space complexity is not that appealing, the paper is still good considering its theoretical contributions. I will increase my score.

Correctness: As mentioned before, the claim in the Abstract about dealing with three different local regularities is a bit confusing. It should state clearly that the algorithm can deal with one of them with different radius tuning function. The proofs seem correct to me.

Clarity: The paper's structure is a bit confusing. It omitted so many things in the main pages that I will fail to understand the ideas and even algorithms if I didn't examine the Appendix carefully. For example, it has Algorithm 1 and 2 in the main paper, which is about local Lipschitzness case. For example, in Algorithm 1, it omitted the update for inactive experts after line 10. In Algorithm 2, it omitted the detail on how to do the prediction for each active expert, which is pretty important in my opinion. What's more, the other two cases' algorithms are only shown in the Appendix. Without the algorithms, it is very difficult for the readers to have a rough intuition on whether your claims are correct. Plus, the local online predictor for each node is using Follow-the-Leader rule. It kind of makes me confused on how to do that with only one node information. The author/s need to explain it better.

Relation to Prior Work: Yes.

Reproducibility: Yes

Additional Feedback: Please see my comments and questions in previous sections.


Review 3

Summary and Contributions: The paper studies different types of 'local' adaptivity in the setting of nonparametric online learning. In particular, the authors present ways to adapt to three different local measures; local Lipschitzness of the comparator function; local metric dimension of the instance sequence; and the base predictors' local performance across the instance space. The techniques are analyzed for the cases of least-squares regression and classification with absolute loss.

Strengths: Using a growing hierarchy of nets on the instance sequence and then applying sleeping experts' techniques to achieve various types of local adaptivity is a very nice approach. The obtained regret bounds can be much better than those due to Hazan and Megiddo 2007. The results for the regression case rely on the exp-concavity of the square-loss. To extend the results to classification with absolute loss (which is not exp-concave), the authors use a parameter-free algorithm, which allows them to avoid parameter tuning. In summary, the paper nicely combines existing (and new) tricks to achieve the desired local adaptivity results.

Weaknesses: The techniques are currently only analyzed for regression with square loss and classification with absolute loss. It would have been nice if the algorithms can be applied beyond these two settings. The algorithms of the paper, though local-adaptive, do not achieve the minimax rate as does the algorithm of Cesa-Bianchi et al 2017. I think the paper would have been more fun to read if more detailed proof sketches were present in the main bound. This would mainly be to briefly showcase how the sleeping expert idea works, or how avoiding the parameter tuning was made possible via the parameter-free algorithm. ----Post Rebuttal---- Thank you for your comments regarding the loss type.

Correctness: The approach makes sense, but I have not checked all the proofs in detail.

Clarity: Overall clear. I found the related works section somewhat oddly placed. I would either move it to the end or just before section 2. I think this may improve the reading flow.

Relation to Prior Work: The related work is discussed thoroughly enough.

Reproducibility: Yes

Additional Feedback:


Review 4

Summary and Contributions: The authors of this paper deal with nonparametric online learning, a setting where regret is measured against rich classes of comparator functions. Finding efficient algorithms that can compete with such complex classes is still an open problem. Therefore, the paper presents a novel algorithm that dynamically grow hierarchical epsilon-nets of the instance space Using a technique based on tree experts, the algorithm simultaneously and efficiently compete against "tree experts". Theoretical guarantee is presented

Strengths: The paper is dealing with and interesting open problem of finding an efficient algorithm for non-parametric online-learning. The theoretical analysis is interesting and as far as I know novel.

Weaknesses: I do except such papers to demonstrate the usefulness of the presented algorithm in the main paper as it gives more insight and intuition to the reader.

Correctness: I didn't find any flaws.

Clarity: The paper is clear.

Relation to Prior Work: Yes, although the authors could have presented the existing algorithms more clearly.

Reproducibility: Yes

Additional Feedback:

[Author Response · NeurIPS 2020]

We would like to thank all reviewers for their careful reviews and insightful comments.

**Reviewer 2: It seems that the algorithm needs to have a pretty large tree. Will it increase in exponential order of T or other order?**

The space grows only linearly in time because the algorithm constructs the tree dynamically, adding a new path of size $\mathcal{O}(D)$ in each round. In particular, the algorithm never allocates the entire tree, but only the paths corresponding to active experts.

**The algorithm can only adapt to one of the mentioned local regularities, and it really just depends on which radius tuning function is used.**

This is correct, and we will modify the abstract to remove any ambiguity. Still, we believe that being able to adapt to different local regularities just by supplying a specific radius tuning function is interesting. Note also that, even though we are not aware of any online nonparametric algorithms with simultaneous adaptivity (not even to global regularities), it is certainly possible to combine algorithms with different types of adaptivity using online aggregation techniques such as prediction with expert advice.

**The paper omitted many details, which can be found in the appendix. In the Algorithm 1, the paper omitted the update for inactive experts after line 10.**

This is not an omission: the algorithm does not require any updates of inactive experts, only experts along the path are updated, which makes it computationally attractive. This is one of the beauties of the context tree weighting method, originally described in the information theory literature (see [29]).

We apologize that some important material had to be placed in the appendix, we will certainly bring back the important details you mention to the main body in the revised version of the paper.

**The local online predictor for each node is using Follow-the-Leader rule. It kind of makes me confused on how to do that with only one node information.**

It would have been more appropriate to call it "follow-the-local-leader". Indeed, each follow-the-leader instance hosted in a ball learns only over the subsequence of examples that fall in that ball.

**Improvement in the local dimension case is doubtful — the oblivious adversary can generate the instance sequence that does not have lower local dimension.**

Note that our algorithms are deterministic (no distinction between oblivious and nonoblivious adversaries), and our regret bounds hold for any sequence. We do not claim that our bound is always better. In fact, when the local dimension equals the global dimension, we merely do not lose anything in the exponent as our bound becomes of order $\widetilde{O}(T^{\frac{d}{1+d}})$.

**The paper assumes that the instance space is the unit ball. What about the case when the instance space is not necessarily a unit ball? Can the algorithm still work in this case? What changes should we make?**

We gain a factor of $C^d$ in the regret (note that this is not improvable), where $C$ is the radius of an instance space. At the same time we have to know $C$ since it will appear in the radius tuning function (see, e.g. [13]).

**Reviewer 1: It seems that we need to specify the hierarchical levels (e.g Lipschitz constants, metric dimensionality) which sound non-trivial.**

The hierarchical levels are hyperparameters determining the reference or comparator class of the nonparametric problem (which in turn defines the inductive bias). Note that the algorithm and its analysis work for *any* choice of these hyperparameters. By choosing specific values, the user controls the trade-off between size of the reference class on one side and the growth of the regret bound and space/time requirements on the other side.

**It is not entirely clear whether the additional (algorithm) complexity and input parameters justify the gains, and if so, on what problems.**

Note that the overhead in running time is logarithmic in the size of the tree, since at each round we only query and update experts along the path of an instance. As for the space requirements, the tree grows only linearly in time as the algorithm constructs the tree dynamically (see response to R2).

**Reviewer 3: The techniques are currently only analyzed for regression with square loss and classification with absolute loss. It would have been nice if the algorithms can be applied beyond these two settings.**

Our proofs can be easily extended to any exp-concave losses, as these do not require any tuning of learning rates in local FTL predictors. We also believe that it is also possible to extend our approach to any convex loss by using a parameter-free local learner such as Squint (Koolen and van Erven, COLT 2015).

[Meta-Review · NeurIPS 2020]

There were four reviews, of which one was initially proposing rejecting and three accepting. The submission addresses an important direction for the development of online algorithms, uses an interesting mix of techniques and achieves good bounds, and the setting is fairly general. Some concerns of the reviewers, such as those regarding running time and hyperparameters, were answered by the authors to the reviewers' satisfaction. However, an issue that came up in discussing the authors' reply was the space complexity, which is O(T). While this is not too bad (even for an online algorithm) and even an inefficient algorithm would serve as a basis for further development, this is still a somewhat negative factor in the decision. In the end, the reviewers reached a consensus for accepting the paper.